# Observation of phonon Poiseuille flow in isotopically purified graphite ribbons

Xin Huang[1,6], Yangyu Guo[1,6], Yunhui Wu [1], Satoru Masubuchi [1], Kenji Watanabe [2], Takashi Taniguchi [1,3], Zhongwei Zhang[1], Sebastian Volz [1,4], Tomoki Machida [1] & Masahiro Nomura [1,5] ✉

In recent times, the unique collective transport physics of phonon hydrodynamics motivates theoreticians and experimentalists to explore it in micro- and nanoscale and at elevated temperatures. Graphitic materials have been predicted to facilitate hydrodynamic heat transport with their intrinsically strong normal scattering. However, owing to the experimental difficulties and vague theoretical understanding, the observation of phonon Poiseuille flow in graphitic systems remains challenging. In this study, based on a microscale experimental platform and the pertinent occurrence criterion in anisotropic solids, we demonstrate the existence of the phonon Poiseuille flow in a 5.5 μm-wide, suspended and isotopically purified graphite ribbon up to a temperature of 90 K. Our observation is well supported by our theoretical model based on a kinetic theory with fully first-principles inputs. Thus, this study paves the way for deeper insight into phonon hydrodynamics and cutting-edge heat manipulating applications.

The classical Fourier's law well describes the diffusive phonon transport in macroscale materials at high temperatures, where the frequent Umklapp phonon-phonon scatterings damp the heat flux. Cooling or down-scaling of the systems invalidates the Fourier's law and gives rise to non-Fourier heat transport behaviors[1–4], such as coherent[5–8], ballistic[9–11], and hydrodynamic[12–16] transport. In contrast to ballistic or coherent phonon transport dictated by the boundary and interface, hydrodynamic transport is governed by intrinsically momentum-conserving normal phonon-phonon scattering. The frequent normal processes lead to exceptionally collective behaviors of phonons similar to those of fluids, including second sound in transient-state[14,15] and phonon Poiseuille flow in steady-state[16,17]. The theoretical prediction and experimental observation of phonon hydrodynamics in solids are of vital significance for both the fundamentals of lattice dynamics due to its unusual physics and the potential applications in thermal management due to its excellent transport properties.

The second sound, named analogously to the first sound (pressure wave), denotes the temperature wave propagating in solid-state materials[18,19]. The phonon Poiseuille flow is similar to that of viscous fluids under the pressure gradient in a pipe. The Poiseuille flow of phonons results from the interplay between normal scattering and diffuse boundary scattering events in the structure with a finite width. Phonon momenta are transferred along the gradient of drift velocity from the sample center to the sides by normal processes and destroyed at the boundaries[13,20,21], inducing a parabolic heat flux profile (Fig. 1a). The experimental detection of second sound in solids has a long history and has been widely reported owing to its direct wavy feature. The drifting second sound was first observed unambiguously in solid He[4] crystals[22], later in various other crystals[23–28] with heat-pulse and light-scattering methods at low temperatures, whereas the driftless counterpart has been detected very recently in Ge even at room temperature under a rapidly varying temperature field[19]. However, owing to the difficulty in observation and lack of direct evidence, there are limited experimental reports on the phonon Poiseuille flow[16,29,30]. Furthermore, this is also partially caused by the ambiguous criterion to confirm the evidence of phonon Poiseuille flow, as to be shown in the present study.

[1]Institute of Industrial Science, The University of Tokyo, Tokyo 153-8505, Japan. [2]Research Center for Functional Materials, National Institute for Materials Science, Tsukuba 305-0044, Japan. [3]International Center for Materials Nanoarchitectonics, National Institute for Materials Science, Tsukuba 305-0044, Japan. [4]LIMMS, CNRS-IIS IRL 2820, The University of Tokyo, Tokyo 153-8505, Japan. [5]Research Center for Advanced Science and Technology, The University of Tokyo, Tokyo 153-0041, Japan. [6]These authors contributed equally: Xin Huang, Yangyu Guo. ✉e-mail: nomura@iis.u-tokyo.ac.jp

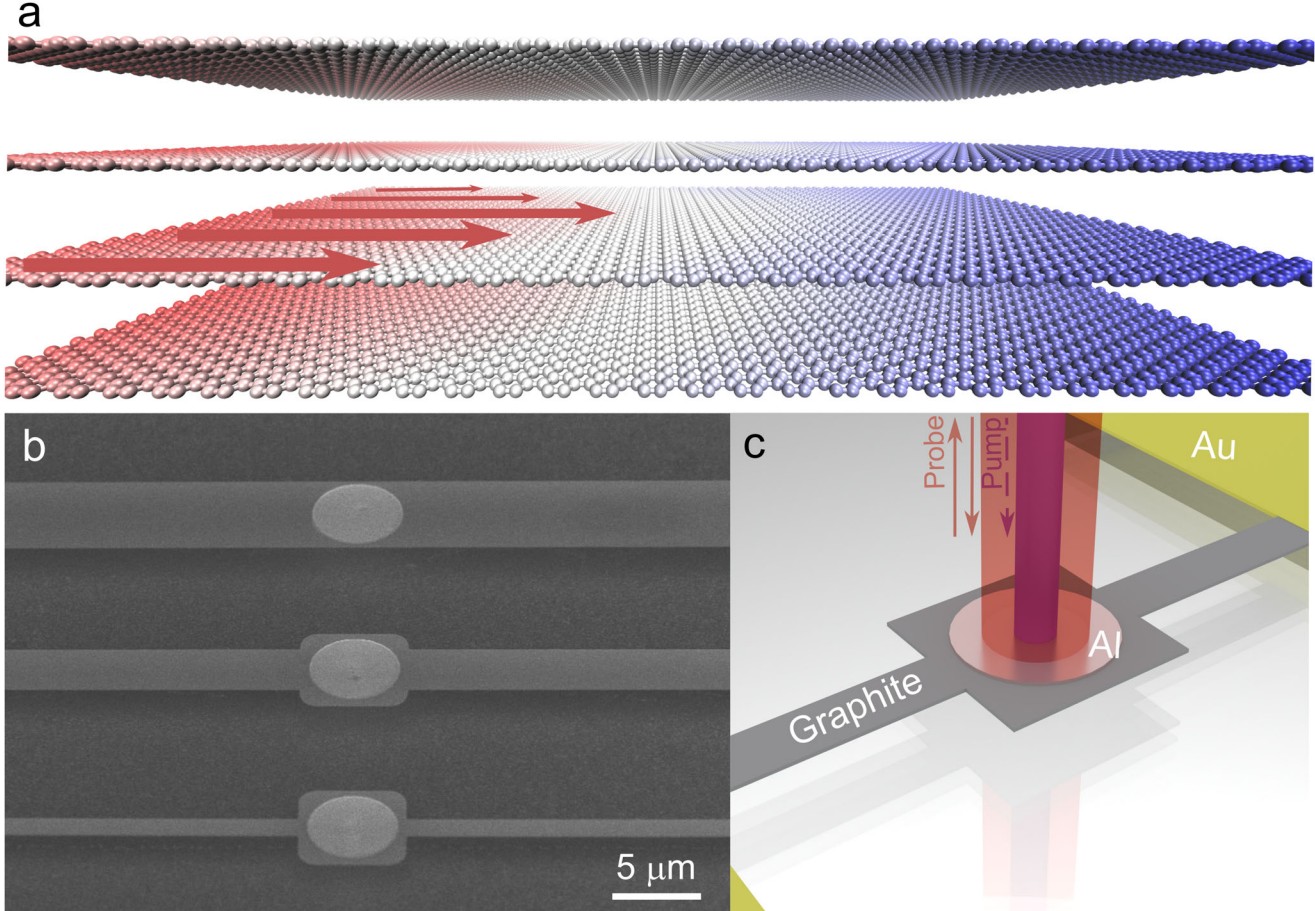

**Fig. 1 | Isotopically purified graphite ribbons and measurement method.**
**a** Illustration of phonon Poiseuille flow in graphite ribbon. In the hydrodynamic regime, the heat flux (represented by the red arrows) manifests a parabolic profile maintained by the collective motion of phonons through a ribbon structure with a finite width. **b** SEM image of suspended graphite ribbons with various widths. **c** Schematic of the μ-TDTR measurement with the pump-probe method.

Graphitic materials, owing to their intensive normal scattering due to the strong anharmonicity, and the high density of states of the low-lying flexural (or bending) phonon modes, are considered as the suitable systems for demonstrating phonon hydrodynamics at elevated temperatures[13,20,21]. The second sound has been recently observed in highly oriented pyrolytic graphite (HOPG) using transient thermal measurement techniques at recording high temperatures[14,15,31]. Despite the numerous theoretical investigations of phonon Poiseuille flow in graphitic materials[17,20,32–34], the experimental observation remains challenging owing to its more stringent observation window condition compared to that of the second sound, as to be elucidated in this work. It requires a special temperature range to realise the dominance of normal scattering and well-designed suspended microstructures to establish the hydrodynamic phonon flow. The indication of phonon Poiseuille flow was reported in a recent experimental work on bulk-scale natural graphite samples[35]. However, it still remains inconclusive due to the pending theoretical explanation of the anomalous thickness-dependent trend and the ambiguous criterion. Additionally, the isotope-phonon scattering, as a momentum-destroying process, has been predicted to play an indispensable role in suppressing the occurrence of phonon Poiseuille flow[12,13]. However, the impact of isotope content in graphitic samples on the phonon hydrodynamic phenomena remains experimentally unexplored.

In this work, we present an unambiguous experimental evidence of phonon Poiseuille flow in graphitic materials. We design and fabricate submicroscale-suspended graphite ribbons and measure the thermal conductivity using a non-contact microsecond-scale time-domain thermoreflectance (μ-TDTR) technique. In addition, we investigate hydrodynamic phonon transport in both the natural and isotopically purified graphite samples in a wide temperature range of 10−300 K. Supported by our first-principles-based theoretical modeling, we uncover the impact of the anisotropic nature of graphite on the criterion of phonon Poiseuille flow, and the appreciable influence of isotope content on its occurrence.

## Results
### Samples and thermal conductivity measurement
Our isotopically purified graphite crystals are synthesised using the high-pressure/high-temperature (HPHT) technique[36,37], and the isotopic abundance of $^{13}C$ is measured to be 0.02% using a time-of-flight secondary ion mass spectrometry (TOF-SIMS) (Supplementary Note 1 and Supplementary Fig. 1a). The $^{13}C$ isotope concentration in the natural graphite crystal is 1.1%. A Raman spectroscopy is employed to characterise the crystallinity of the sample. As seen in Supplementary Fig. 1b, the Raman spectra show two main peaks for both crystals: the G peak at ~1561 cm$^{-1}$ and the 2D peak at ~2710 cm$^{-1}$, representing the sp$^2$ bonding of carbon atoms and perfect crystallite of the samples[38–40]. The only difference between these two samples is the isotopic concentration of $^{13}C$. Both samples are treated under the same conditions in the entire fabrication and measurement process. Starting from a ~50 × 150 μm$^2$ graphite flake, we first pattern and fabricate several suspended graphite ribbons connecting the graphite islands and the gold heat sinks from both sides (Fig. 1b). All the ribbons have the same length of 30 μm, and various widths from 1.3 to 5.5 μm. Next, we apply the pump-probe technique for investigating the in-plane phonon transport through the graphite ribbons with a circular aluminum

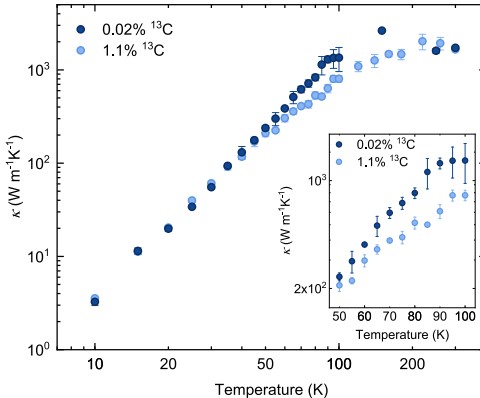

**Fig. 2 | Temperature-dependent in-plane thermal conductivity ($\kappa$) of isotopically purified and natural graphite ribbons with a designed width of 5.5 μm.** The dark and light blue dots represent the results of isotopically purified (0.02% $^{13}$C) and natural (1.1% $^{13}$C) graphite ribbons, respectively. Note that the actual width of the natural graphite ribbon is 0.8 μm wider than that of the isotopically purified one due to the deviation in fabrication, resulting in the minor flip of thermal conductivities at very low temperatures. Inset: thermal conductivity from 50 to 100 K. Error bars depict the standard deviations of different measurements on the same ribbon.

transducer deposited on the central island, as depicted in Fig. 1c. Our μ-TDTR setup ensures the precise measurement of the steady-state thermal conduction in the ribbons at the microsecond-scale (see details in Methods). The thickness of the ribbons is identical to that of the initial flake, and it is estimated to be ~85 nm from the cross-section view of the scanning electron microscope (SEM) image (Supplementary Fig. 1a). Other sample conditions, including the impurity concentration and surface roughness, are assumed to be consistent for all the adjacent ribbons from the same flake to justify the width-dependent investigation in the following content.

Figure 2 shows the in-plane thermal conductivity of the isotopically purified (0.02% $^{13}$C) and natural (1.1% $^{13}$C) graphite ribbons with the same designed width of 5.5 μm. We measure the thermal conductivity of both the ribbons from 300 K down to 10 K, which sufficiently covers the temperature range of hydrodynamic window condition in graphite as reported in previous theoretical and experimental studies[14,17,35]. With the decrease in temperature from 300 K, as the Umklapp phonon scattering becomes weaker, the thermal conductivities of both samples follow an increasing trend until they reach their peaks. The peak value of natural sample is measured as ~1477 Wm$^{-1}$K$^{-1}$, which is lower than that of bulk HOPG with the same isotope contents[35] reported in a recent work owing to the strong size effect from structure down-scaling[41–43] (Supplementary Fig. 2). While the thermal conductivity of isotopically purified sample is peaked at around 150 K as ~2635 Wm$^{-1}$K$^{-1}$ owing to the isotopic enrichment. The phonon Umklapp process is further weakened below 150 K, where appreciable difference emerges between the thermal conductivities of the two samples with different $^{13}$C concentrations. The isotope scattering plays an important role in this regime, resulting in an enormous enhancement of thermal conductivity of isotopically purified graphite ribbon compared to that of the natural one by 105% at 90 K, which also qualitatively affects the occurrence of phonon Poiseuille flow as discussed in the following content. The isotopic effect remains prominent until the temperature goes down to 50 K, below which the thermal conductivities of the two samples become comparable again due to the gradual dominance of boundary scattering over isotope scattering. These trends of the thermal conductivities are similar in natural and isotopically purified graphite ribbons with the widths of 1.3 μm and 3.3 μm, as shown in Supplementary Fig. 3. The effect of isotopic

enrichment on the increase of thermal conductivity has also been observed in many other materials within the temperature range of 128−380 K, such as GaN (15%)[44], boron phosphide (17%)[45], graphene (36%)[46], diamond (50%)[47], cubic boron nitride (90%)[48], and Si NWs (150%)[49].

## Phonon Poiseuille flow

In the hydrodynamic regime, the collective motion of thermal phonons due to the dominant intrinsic normal process demonstrates a Poiseuille flow of heat, as shown in Fig. 1a. In this regime, the probability of phonons losing momentum (due to resistive scattering) is notably reduced along their transport paths. In contrast, the phonon momentum is frequently destroyed in the ballistic regime due to the diffuse boundary-phonon scattering events. Therefore, a faster increase in thermal conductivity than the ballistic limit is considered as the indicator of the phonon Poiseuille flow[12,13,20]. Quantitatively, a simple kinetic formula estimates the thermal conductivity as $\kappa \sim Cvl$, with $C$, $v$, and $l$ being the heat capacity, group velocity, and mean free path (MFP), respectively. The effective momentum-destroying MFP in the hydrodynamic transport can be obtained from the random walk theory as[50,51]: $l \sim W^2/l_N$, with $W$ as the sample width and $l_N$ as the MFP of normal process. One may feature the hydrodynamic thermal transport in principle from the temperature-dependence of $l$ associated with the strength of the normal process, as detected in some crystals[30,52]. The MFP is instead limited by the sample width ($l \sim W$) in the ballistic limit: thus, $\kappa \sim CvW$. For most common three-dimensional (3D) materials, in the hydrodynamic regime, meaning that at very low temperatures, the heat capacity follows the well-known Debye T$^3$ law, and the group velocity is approximately the speed of sound as a constant. Therefore, as the temperature increases, with the enhancement of normal scattering ($l_N$ decreases), the thermal conductivity also boosts more rapidly than the ballistic limit (T$^3$). As a result, the increase in thermal conductivity, with a temperature-dependent exponent larger than 3, has been adopted as a criterion to confirm the hydrodynamic phonon flow in several 3D crystals, such as SrTiO$_3$ (T$^{-3.5}$)[16], Bi (T$^{3.5}$)[52], and He$^4$ (T$^8$)[53], within the temperature ranges of 6−13 K, 1.5−2.4 K, and 0.7−0.9 K, respectively. For a two-dimensional (2D) system like graphene, the ballistic thermal conductance (or conductivity) follows T$^{1.68}$[41], and a faster increasing trend of thermal conductivity over T$^{1.68}$ is used to indicate the phonon Poiseuille flow, and it has been obtained by varying the width of graphene ribbon in a previous theoretical study[20].

However, the situation becomes quite different for graphite, where single graphene layers are bonded through weak van der Waals force. The anisotropic nature of graphite makes the temperature scaling of the thermal properties different from those of its 2D counterpart (graphene) and other isotropic 3D materials. The measured heat capacity of graphite deviates from the Debye law and shows a smaller power dependence of T$^{2.5}$ with an exponent between those of 2D and 3D systems[54]. In a recent experimental report with natural HOPG sample, an increase in the ratio of thermal conductivity over T$^{2.5}$ or over heat capacity with increasing temperature has been adopted to indicate the phonon hydrodynamics[35]. In the following part, we will illustrate that the ballistic limit (or ballistic thermal conductance ($G_{ballistic}$), equivalently) of graphite shows a different temperature dependence from that of heat capacity, i.e., different from T$^{2.5}$. Therefore, a faster increase of $\kappa$ than $G_{ballistic}$ with increasing temperature is expected for the presence of phonon Poiseuille flow in graphite. Thus, we will demonstrate unambiguously the evidence of phonon Poiseuille flow in isotopically purified graphite ribbons with sufficiently large widths.

The results of $\kappa/T^{2.5}$ and $\kappa/G_{ballistic}$ as a function of temperature are given in Fig. 3a and b, respectively, for our isotopically purified graphite ribbons. The ballistic thermal conductance for graphite is calculated by the first-principles method (see details in Methods) as

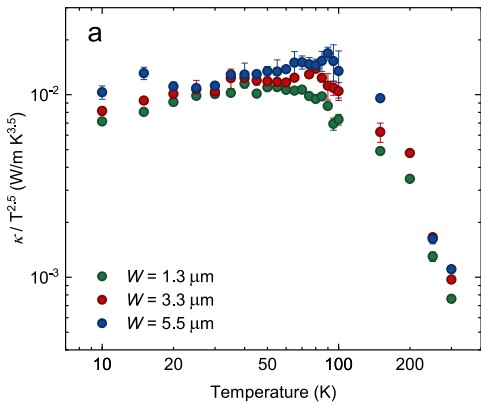
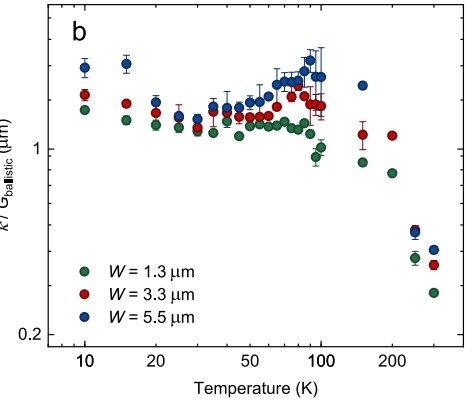

**Fig. 3 | The criterion and evidence of phonon Poiseuille flow in isotopically purified graphite ribbons. a** The usual criterion, namely the ratio of thermal conductivity ($\kappa$) over $T^{2.5}$ as a function of temperature (T) for graphite ribbons with various widths (W). **b** The present criterion, namely the ratio of thermal conductivity over $G_{\text{ballistic}}$ as a function of temperature. Error bars depict the standard deviations of different measurements on the same ribbon. The thermal conductivities of the isotopically purified graphite ribbons are shown here.

follow:

$$G_{\text{ballistic}} = \sum_p \int v(\mathbf{k}) \hbar \omega(\mathbf{k}) \frac{\partial f^{eq}}{\partial T} \frac{d\mathbf{k}}{(2\pi)^3}, \qquad (1)$$

where $p$ represents phonon polarization, and $v(\mathbf{k})$, $\mathbf{k}$, $\hbar$, $\omega(\mathbf{k})$, and $f^{eq}$ are the group velocity, wave vector, reduced Planck's constant, frequency, and Bose-Einstein equilibrium phonon distribution, respectively. Note that we also calculate the ballistic thermal conductance based on an empirical atomic interaction potential. The quantitative value of the ballistic thermal conductance will be influenced by the atomic interaction potential. However, the qualitative temperature-scaling behavior and the conclusion will be not much changed. More detailed discussions are given in Supplementary Note 2 and Supplementary Fig. 4.

As shown in Fig. 3a, $\kappa/T^{2.5}$ increases with temperature from 10 to ~40 K for all the isotopically purified graphite ribbons with different widths, including in the narrowest case of 1.3 µm, where momentum-destroying boundary scattering is expected to remain appreciable. We further examine the temperature dependence of $\kappa/T^{2.5}$ in a 500 nm-wide ribbon, where the heat transport should lie within the ballistic regime. However, an unexpected raise of $\kappa/T^{2.5}$ as temperature increases is observed, as shown in Supplementary Fig. 5a. This could be explained by the faster increase of the $G_{\text{ballistic}}$ than $T^{2.5}$ in the same temperature range, as illustrated in the inset of Supplementary Fig. 5a. In other words, a faster increase of $\kappa$ than $T^{2.5}$ or the heat capacity may not indicate the occurrence of hydrodynamic phonon flow definitely. Thus, a more relevant criterion to demonstrate phonon Poiseuille flow in graphite would be the temperature-dependent trend of $\kappa/G_{\text{ballistic}}$, as shown in Fig. 3b. In the graphite ribbon with a width of 1.3 µm, $\kappa/G_{\text{ballistic}}$ continuously decreases with increasing temperature, as a sign of the transition from ballistic to diffusive transport. The trend is similar in the case of 500 nm-wide ribbon, as indicated in Supplementary Fig. 5b. In these cases, the sample widths are too narrow for enough normal scatterings to occur. However, in the graphite ribbon with a larger width of 3.3 µm, $\kappa/G_{\text{ballistic}}$ starts to increase with increasing temperature from 50 to 80 K, where the normal scattering starts to play an increasing role to cancel and prevail the effect of momentum-destroying phonon scatterings. When the width of the graphite ribbon is sufficiently large as 5.5 µm, the normal scattering becomes frequent while the resistive (Umklapp and isotope) scattering is still scarce, and an apparent enhancement of $\kappa/G_{\text{ballistic}}$ is observed from 40 K. The temperature window of phonon Poiseuille flow is expanded and lasts to an elevated upper limit of 90 K owing to the

dominance of momentum-conserving normal scattering in the ribbon with larger width. This super-ballistic scaling of thermal conductivity with temperature is a clear evidence of phonon Poiseuille flow. At higher temperatures (>100 K), $\kappa/G_{\text{ballistic}}$ shows a dramatic decrease with increasing temperature in the graphite ribbons with all the widths due to the increasing rate of Umklapp scattering.

To observe the phonon Poiseuille flow, the rate of normal scattering should be dominant over those of the resistive ones, such as Umklapp scattering and phonon-isotope scattering[12,13]. To this end, we also investigate the isotope effect on the phonon Poiseuille flow by comparing the results of isotopically purified (0.02% $^{13}$C) and natural (1.1% $^{13}$C) graphite ribbons. We first examine the occurrence of phonon Poiseuille flow based on the temperature dependence of $\kappa/G_{\text{ballistic}}$ in 1.3 µm-wide graphite ribbons (Fig. 4a). As illustrated in Fig. 4d, $\kappa/G_{\text{ballistic}}$ of both samples show a decreasing trend as the temperature increases due to the predominant diffuse phonon-boundary scattering in relatively narrow ribbons. A steeper decrease is found in the natural graphite sample due to the additional effect of the momentum-destroying isotope scattering of phonons. When the ribbon width is sufficiently large, namely the 3.3 µm case (Fig. 4b), $\kappa/G_{\text{ballistic}}$ shows a non-monotonous trend in the isotopically purified sample, while it continuously decreases in the natural one with increasing temperature, as shown in Fig. 4e. It infers that the phonon Poiseuille flow is deteriorated by the resistive phonon-isotope scattering in natural graphite sample. Further enlargement of the ribbon width to 5.5 µm (Fig. 4c) makes the difference of $\kappa/G_{\text{ballistic}}$ between isotopically purified and natural samples more pronounced, as depicted in Fig. 4f. This is caused by the larger space for sufficient momentum-conserving normal scattering to occur in the purified sample. On the opposite, widening of the ribbon also makes more momentum-destroying phonon-isotope scattering to occur in the natural counterpart.

The aforementioned tendencies of experimental data are generally consistent with our theoretical modeling results in Fig. 4g–i based on a direct solution of phonon Boltzmann transport equation (BTE) with full first-principles inputs (see details in Methods). There is some difference between the absolute values of $\kappa/G_{\text{ballistic}}$ in experimental and theoretical results, as we consider infinite thickness in the BTE modeling. However, a good agreement is found in terms of the relative trends of isotopically purified and natural abundance graphite ribbons and the temperatures where the minimum and maximum emerge. A direct solution of phonon BTE for hydrodynamic heat transport in graphite ribbon with finite length, width and thickness is a challenging task, as it requires a numerical solution in both 3D reciprocal space and 3D real space. To the authors' best knowledge, it is

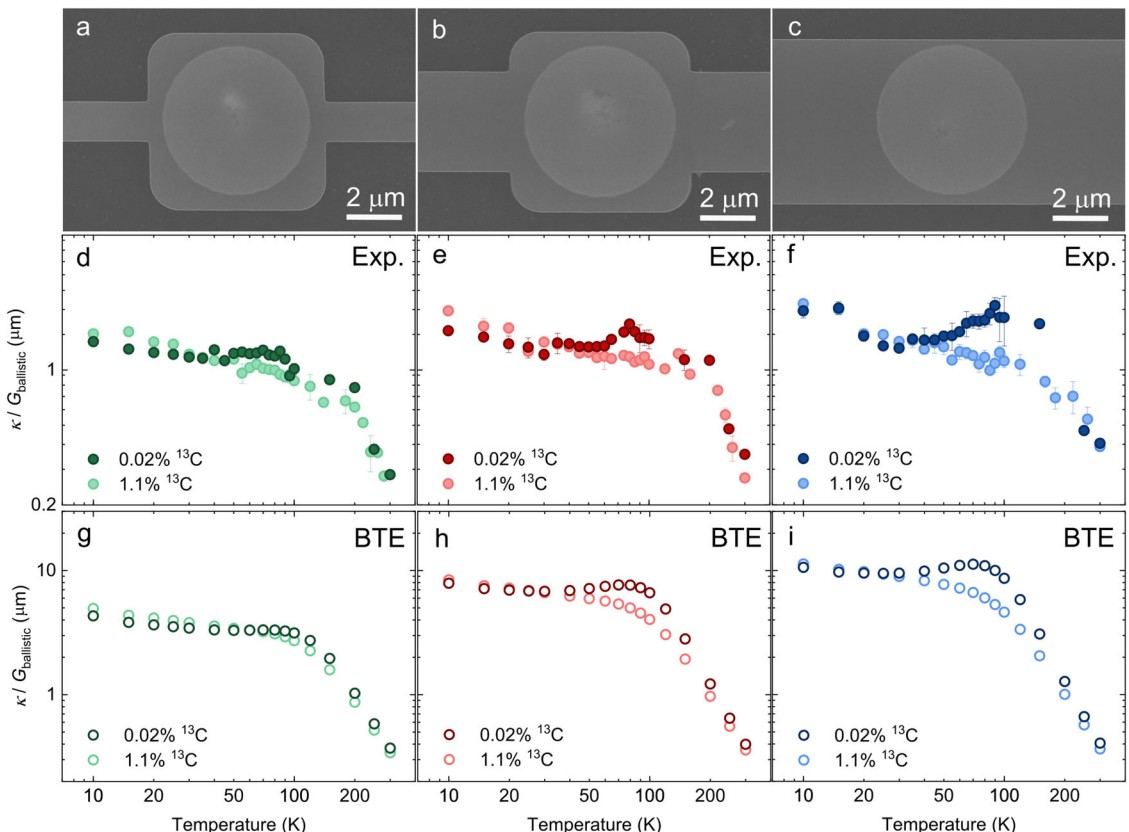

**Fig. 4 | Isotope effect on phonon Poiseuille flow in graphite ribbons. a–c** SEM images of suspended isotopically purified graphite ribbons with the widths of 1.3 μm, 3.3 μm and 5.5 μm, respectively. **d–f** Experimentally measured and (**g–i**) calculated thermal conductivity ($\kappa$) over ballistic thermal conductance ($G_{ballistic}$) as a function of temperature corresponding to the three ribbons in (**a–c**). The dark (light) green, red (pink), and dark (light) blue dots represent the experimental data of isotopically purified (natural) graphite ribbons. Error bars depict the standard deviations of different measurements on the same ribbon. The empty dots with the corresponding colors denote the modeling results by BTE with first-principles inputs. Note that the actual widths of the natural graphite ribbons are 1.6 μm, 3.7 μm and 6.3 μm, respectively, due to the deviation in fabrication, resulting in the minor flip of $\kappa/G_{ballistic}$ at very low temperatures.

only reported that the Monte Carlo solution of phonon BTE with *ab initio* full scattering term for such situation from one group in very recent studies[15,55]. However, due to huge computational cost, relatively coarse grids in both reciprocal and real spaces have been adopted. Apparently, there is still some space to further improve the accuracy of the numerical solution and its agreement with experimental result[15]. On the other hand, as the thickness effect on basal-plane heat transport in graphite remains an open question[35,55], the present modeling and experimental study are mainly focused on the effects of finite length and width. Our semi-quantitative theoretical modeling generally provides a good guide for the observation of phonon Poiseuille flow in finite-sized isotopically purified graphite ribbons.

## Discussion

The steady-state phonon hydrodynamic phenomenon, i.e., phonon Poiseuille flow, appears only when a strict condition is satisfied. It requires normal scattering to be more sufficient than boundary scattering, which is further more substantial than other resistive phonon scattering events, such as Umklapp and isotope scatterings. In nanoscale structures, the frequent interaction between phonons and the structure edges brings heat conduction to the ballistic regime. In much larger and isotopically-impure samples, the resistive scattering deteriorates the hydrodynamic phonon flow. Thus, the Poiseuille flow of phonons can be well-established in the graphite sample with a purified isotope concentration and a width in between the MFPs of normal and resistive scatterings[56] ($l_N \ll W$, $l_R l_N \gg W^2$). A recent theoretical work predicted that the width window condition of phonon

hydrodynamics is approximately $2-20$ μm in the temperature range of $50-90$ K in graphite ribbons with 0.1% isotope content[17]. Our observation of phonon Poiseuille flow in 3.3 μm- and 5.5 μm-wide isotopically purified graphite ribbons, therefore, confirms this theoretical prediction. We also provide a detailed quantitative demonstration of why the hydrodynamic window condition is satisfied only in the isotopically purified graphite ribbon in Supplementary Note 2 and Supplementary Fig. 6. On the other hand, it explicitly demonstrates a more stringent condition for the observation of phonon Poiseuille flow than that of the second sound, which has been observed instead in natural graphite recently[14,15,31] (detailed explanation in Supplementary Note 3).

We have shown that the temperature dependence of $\kappa/G_{ballistic}$ (or equivalently $\kappa/\kappa_{ballistic}$) is a more relevant criterion to confirm the phonon Poiseuille flow. In most 3D materials close to isotropic structures, the ballistic thermal conductance and heat capacity follow the same temperature power law at low temperatures due to the linear dispersion relation of acoustic phonons[20,57,58]. The different temperature scalings between $G_{ballistic}$ and heat capacity ($T^{2.5}$) of graphite, shown in the inset of Supplementary Fig. 5a, is mainly attributed to the anisotropic nature and special phonon dispersion of graphite. The hydrodynamic phonon transport is mainly contributed by the bending acoustic (BA) modes in graphite[17,33], the group velocity of which increases with frequency due to the quadratic dispersion curve[13,59]. As expressed in equation (1), $G_{ballistic}$ is determined by both the heat capacity term and the group velocity term ($v(\mathbf{k})$). Therefore, with an increase in temperature in the hydrodynamic window and more populated higher-frequency BA phonons, $G_{ballistic}$ of graphite boosts

faster than the heat capacity (T$^{2.5}$). As a result, the increase of $\kappa/$T$^{2.5}$ with temperature from few Kelvins to ~25 K in the recent experimental report[35] is most probably contributed by the behavior of the ballistic thermal conductance.

In addition, as a reference, we examine the isotope effect on the observation of phonon Poiseuille flow based on the present criterion ($\kappa/G_{\text{ballistic}}$) in silicon samples with natural and purified $^{28}$Si isotope concentrations[60] (Supplementary Fig. 7). As temperature increases, $\kappa/G_{\text{ballistic}}$ of both natural and purified silicon samples drops monotonically, indicating the absence of phonon Poiseuille flow even in the isotopically purified silicon sample. This is explained by the well-known insufficient normal scattering in silicon to satisfy the hydrodynamic window condition.

According to the different temperature-dependent behaviors of $\kappa/G_{\text{ballistic}}$ in our isotopically purified graphite ribbons with various widths, we observe the transition from the ballistic to the hydrodynamic thermal transport when the ribbon width increases from 1.3 μm to 5.5 μm. Besides, another important aspect to evidence the phonon hydrodynamic flow is the super-ballistic width dependence of thermal conductivity, as clearly indicated by the effective mean free path $l \sim W^2/l_N$. The suspended microstructure system built up in this work provides a good platform for further experiments to investigate the extraordinary super-linear width dependence of thermal conduction in the hydrodynamic regime or the phonon Knudsen minimum phenomenon[17,32,33,61].

Finally, we would like to note that the phonon Poiseuille flow could be modeled by macroscopic phonon hydrodynamic equations[2,56] in a much more efficient way. It resembles the description of Poiseuille flow of classical fluids by the Navier-Stokes equation. The classical Guyer-Krumhansl (G-K) phonon hydrodynamic equation[56] assumes gray phonon properties and usually works well for traditional 3D isotropic crystals at very low temperatures. However, the complex nonlinear frequency-dependent phonon properties are important for phonon hydrodynamics in graphitic materials where the original G-K equation might be not able to be directly applied. Recently, a generalized G-K heat transport equation (so-called kinetic-collective model, KCM[62]) has been derived from phonon BTE taking into account the arbitrary scattering term and the nonlinear phonon properties[63]. In principle, the KCM, together with appropriate macroscopic boundary conditions, could be an alternative for modeling the phonon Poiseuille flow in finite-sized graphite ribbon, which is however a nontrivial task and beyond the scope of the present study. On the other hand, the hydrodynamic model (or KCM) covers the heat transport in both kinetic[62,64] and collective[65] limits. Apparently the debate about the definition of the hydrodynamic regime[14,19] when heat transport could be described by the KCM is still open. Nevertheless, the present work is mainly focused on the collective limit, i.e. when normal scattering dominates. Thus we could not provide definite remarks about this debate in the current stage, and leave it for a future exploration.

In summary, we have developed an integrated experimental platform to investigate the steady-state phonon hydrodynamics in suspended graphite submicron structures. In light of a more relevant criterion due to the anisotropic nature of graphite, we observed prominent Poiseuille flow of phonons in an isotopically purified graphite ribbon with a width of 5.5 μm up to 90 K, which is much elevated compared to the temperature range of previous observations in other solid-state materials[16,30,52,53]. The phonon Poiseuille flow is prone to be destroyed by the resistive phonon-isotope scattering in graphite ribbons with natural abundance of carbon isotope. Our joint theoretical and experimental study on phonon hydrodynamics in graphitic materials thus deepens the understanding of the collective physics of phonons in anisotropic solids. The experimental platform will also open innovative possibilities for tuning and manipulation of phonon hydrodynamics, as well as its application in thermal management of the modern micro- and nanoelectronics.

## Methods

### Sample preparation

Sample fabrication begins with mechanical exfoliation of graphite to obtain graphite flakes, which are then transferred onto a SiO$_2$ (2.4 μm)/Si substrate right after O$_2$ plasma surface treatment. The typical size of a flake is ~50 × 150 μm$^2$, and the flake thickness is approximately 85 nm (as shown in Supplementary Fig. 1a), measured by SEM. Next, we apply electron-beam lithography (EBL) for patterning ribbon structures with a fixed length of 40 μm but varying widths. Furthermore, a 6 × 6 μm$^2$ graphite island is patterned in the center of the ribbons on the same flake. We also use an electron-beam physical vapor deposition (EBPVD) to deposit 100 nm-thick aluminum on top of the graphite ribbons as masks for O$_2$ plasma etching. After exposing samples in a reactive ion etching chamber with an O$_2$ plasma source, we remove the rest of the graphite around the ribbon structures and release the aluminum masks to acquire the desired graphite ribbons. Moreover, we use laser lithography and another EBPVD to fabricate two 250 × 400 μm$^2$ gold pads used as hydrofluoric acid (HF) stoppers, to cover all the ribbons by 10 μm-long from both sides. Following this, 70 nm-thick circular aluminum transducers with radius of 2.5 μm are deposited on the central graphite island for TDTR measurement. HF vapor etching is used to remove SiO$_2$ from the entire surface, with only a portion of SiO$_2$ remaining underneath the gold pads to support the pads. Being attached and clamped by the gold pads from both sides, the ribbons are finally suspended for investigating phonon hydrodynamic transport, with a length of 30 μm staying completely out-of-contact.

### Thermal characterisation

We employ a well-developed μ-TDTR setup to measure the thermal conductivity of our samples[66]. We place our samples in a vacuum cryostat with a pressure below 10$^{-5}$ Pa to avoid the convective heat loss. At 200 K, radiation heat loss from graphite ribbons is estimated as ~1 nW, whereas the input power from the pump beam is ~ 200 nW. At temperatures below 150 K, the loss through radiation is negligible[67,68]. A liquid helium flow system enables to cool down the cryostat to 4 K. A cryogenic temperature controller from Oxford Instruments adjusts the temperature with a precision of 0.1 mK.

In the μ-TDTR setup, two lasers are focused onto the center of the aluminum transducer located on the graphite island: one pulsed pump beam with a wavelength of 642 nm and one continuous probe beam with a wavelength of 785 nm. The pump beam with a 2 μs pulse duration at 1 kHz repetition rate induces excitation in the transducer. The probe beam measures the reflectivity change from the base value generated by the pump beam, and a photoreceiver monitors its reflection with a maximum bandwidth of 200 MHz to detect the response every 5 ns. An oscilloscope captures the probe signal averaging 10$^4$ measurements into a thermal decay as a function of time, as shown in Supplementary Fig. 8. The fitting parameter, $\tau$, is an indicator of the inherent thermal properties of the measured sample. We measure each ribbon with one single-width three to five times and calculated its standard deviation as depicted by the error bars in figures containing the experimental data.

To extract the thermal conductivity of the graphite ribbons, we build up a numerical model identical to the actual experimental one using the finite element method (in COMSOL Multiphysics). In the model, the material parameters of metals (gold and aluminum) are assumed as their bulk values due to neligible size effects, whereas the specific heat of graphite is taken from the reported benchmark data[69]. The anisotropic nature of heat conduction in graphite is taken into account by setting the bulk out-of-plane thermal conductivity from the literature[70] and leaving the in-plane thermal conductivity as the only fitting parameter. Thermal boundary conductance between metals and graphite is also regarded as a crucial parameter in the model to correctly reproduce the experimental measurement, as further explained in details in Supplementary Note 4 and Supplementary

Fig. 9. By injecting a 2 μs heat flux pulse with a Gaussian spatial distribution onto the aluminum transducer, we simulate the heat dissipation through the graphite ribbon structures to reproduce the decay times in the experiments by sweeping the values of in-plane thermal conductivity. The in-plane thermal conductivity of graphite ribbon is obtained when an optimal fitting was found between the decay times obtained by simulation and by experiment (Supplementary Fig. 8).

## Theoretical modeling

We model the hydrodynamic heat transport through finite-size graphite ribbons by directly solving the phonon Boltzmann transport equation (BTE) with fully first-principles inputs using the methodology developed in our recent work[33]. The required input of phonon properties (including phonon dispersion relations and scattering rates) are calculated in the open-source package SHENGBTE[71], with the harmonic and anharmonic (third-order) force constants obtained by the first-principles calculations implemented in the package QUANTUM ESPRESSO[72]. To accurately describe the inter-layer interaction in graphite, we adopt the most advanced van der Waals non-local density functional theory, with all the numerical details and validation of the first-principle calculations provided in our previous work[33]. The first-principles phonon properties are also used in the calculation of the ballistic thermal conductance in equation (1). In the direct numerical solution of phonon BTE, we model the graphite ribbon with the same length, width, and carbon isotope concentration as those in the experimental measurement. Fully diffuse scattering of phonons at the transverse boundaries of the graphite ribbon are considered. This is reasonable as the edges of the ribbon are generally roughened by etching process. Along the cross-plane direction of graphite ribbon, we assume infinite thickness and solve the phonon BTE in 2D real space and 3D reciprocal space[33]. Such treatment is strictly valid when the surfaces in the thickness direction are fully specular, i.e., ideally smooth, which is an acceptable case when the sample is obtained via perfect exfoliation process as in multilayer graphene ribbon[73,74]. Since the absolute thermal conductivity by the present BTE modeling is larger than the experimental one, we infer that there should be unknown surface imperfections in the thickness direction. Nevertheless, the present BTE modeling captures the dominant effects from finite length and width, considering that the numerical solution of BTE in both 3D real and reciprocal spaces is computationally too expensive for hydrodynamic heat transport. Also, our recent numerical methodology considering only finite length and width[33] remains to be further developed, which requires appreciable amount of future work and effort.

## Data availability

The data that support the findings of this study are available from the corresponding authors upon reasonable request.

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

## Acknowledgements

X.H. acknowledges the support from the Grant-in-Aid for JSPS Fellows (Grant Number 21J12652). Y.G. acknowledges the support from the Grant-in-Aid for JSPS Fellows (Grant Number 19F19353). S.M. acknowledges the support from JSPS KAKENHI (Grant Number 19H01820). K.W. acknowledges the support from JSPS KAKENHI (Grant Number 21H05233). T.M. acknowledges the support from JSPS KAKENHI (Grant Numbers 20H00127, 21H05232, 21H05234) and CREST JST (Grant Number JPMJCR20B4). M.N. acknowledges the support from JSPS KAKENHI (Grant Number 21H04635) and CREST JST (Grant Number JPMJCR19Q3).

## Author contributions

X.H., Y.G. and M.N. conceived this study. X.H. and Y.W. designed and conducted the fabrication. X.H. performed the TDTR and Raman spectra measurements, analyzed the results, and wrote the manuscript. S.M. and T.M. provided the graphite samples, mechanical exfoliation technique and Raman spectroscopy technique. S.M. contributed to designing the fabrication and Raman spectra measurements. K.W. and T.T. synthesized the isotopically purified graphite crystals. Y.G. developed the theoretical model and contributed to writing the manuscript. Z.Z. and S.V. contributed to interpreting the results. M.N. supervised this work. All authors contributed to discussing the results and manuscript revision.

## Competing interests

The authors declare no competing interests.
