## [Peer Review File · Nature Communications]

REVIEWER COMMENTS

Reviewer #1 (Remarks to the Author):

The authors claim that they observe a non-monotonous behavior of the effective thermal conductivity of a graphite sample in terms of temperature. They claim that this is evidence of the emergence of a Poiseuille flow.

In the last years phonon hydrodynamics has been proposed to understand the non-Fourier behavior of thermal transport at the nanoscale. This has been very fruitful to understand 2D samples like graphene or samples like graphite that can be understood as a van der Waals stacking of graphene layers. In the last years there has been very intense efforts to experimentally observe it. Poiseuille flow is a phenomenon directly related with the hydrodynamic behavior of heat, and its direct observation has been pursued as evidence in favour of phonon hydrodynamics. These kinds of results are very interesting to the research community as they can help to identify in which situations one can expect hydrodynamics in thermal transport.

Despite the clear importance of the topic, and in particular the experiment that the authors present in their work, I have some criticism that to my opinion should be raised before publishing the work.

a) The evidence in favour of the Poiseuille flow is based in the plots 4d/4e/4f. The authors base their arguments by claiming that there is a clear difference between plot 4f and plot 4d. From their point of view, there is a difference in the slope in the pure sample in 4f respect to that of 4d. Overlapping figure 4d with figures 4e-4f (see image in the attached pdf), it seems that there is no clear evidence in favour of the authors claims:

1) The observation of the peak near 100K it is not evident outside the statistical fluctuation in the measurements. The slope of the pure sample in 4f is not statistically different from the slope in figure 4d, where the authors claim that there is no Poiseuille flow. Even accepting that there is a peak near 60K, there is no explanation of the difference with the peak observed near 100K in the same sample. The authors should explain the

difference between both and give an statistical differentiation between them.

2) The difference between samples with pure and natural isotope concentrations are almost identical for all the samples (4d, 4e, 4f). In the experimental data there is no evidence in favour of the differences predicted by the theoretical model used to interpret the data. In the solid lines (theoretical predictions) there is a clear tendency of the pure sample to merge with the natural sample. Once again, by overlapping plots 4c and 4e-4f this tendency is not clearly observed in the experimental data.

b) Regarding the theoretical framework presented in the paper to identify the appearance of the Poiseuille flow, I have some concerns that to my opinion the authors should discuss.

In the first papers where Poiseuille flow has been observed, the behavior of thermal conductivity is compared to a T^3 curve. If the slope of thermal conductivity is larger than 3 in a log-log plot, Poiseuille flow is inferred. The theoretical explanation for this is clear in an isotropic material. At low temperatures, the mean free path saturates to the width of the sample. The velocity of the phonons is constant because in the low temperature region, the slope in the distribution function does not change. This leaves the specific heat as the only temperature dependent magnitude. This means that the test of the Poiseuille flow can be done in a pure experimental way by comparing the experimental data of thermal conductivity with that of the specific heat. As the slope of specific heat is constant, and only depending on dimensionality for 3D isotropic materials, this test offers no debate.

In 2D materials, this is more subtle. The quadratic band in the dispersion relations has an effect on the specific heat slope and the relation with the dimensionality is broken. This means that without access to experimental data on specific heat, its slope may depend on the theoretical model. The authors suggest that ballistic conductance should be done in order to solve this. That is, a temperature slope larger than the ballistic conductance should be used to identify Poiseuille flow.

Even accepting that the dimensionality cannot be used, the authors should present their results with a brief discussion of this important point. They should study the influence of the variation of the dispersion relations on the theoretical curves of the ballistic behaviour and the impact that may have on the conclusions of the paper.

c) Regarding the novelty of the results, I should say that the observation Poiseuille flow in dielectrics is not new. It was experimentally observed on LiF and NaF in the 70's (<https://doi.org/10.1002/pssb.2220520127>) and very recently, a very similar work has been published in Science Advances (<https://doi.org/10.1126/sciadv.aat3374>) for Black Phosphorous. In this last work, the comparison with the experimental data of specific heat slope is done directly, and not through a ballistic conductance line.

d) In a more general scope, I should mention that in the scientific community there is a very important point of discussion regarding the microscopic justification for the appearance of the hydrodynamic behavior in thermal transport. The authors point of view follows the traditional point of view of the 60's initiated by Pierls and contemporary authors. They argue that the dominance of Normal collisions as the only justification for the appearance of hydrodynamics. But in the last years, there has been a lot of works pointing to behaviours that are not in agreement with that point of view. Two important ones are the observation of second sound in Germanium (<https://doi.org/10.1126/sciadv.abg4677>), and (<https://doi.org/10.1038/s41467-021-27907-z>). The last one is in apparent contradiction with this work, as they observe second sound in natural isotopic graphite over 200K. From the point of view offered in the introduction of this work, these two observations cannot be explained. Despite nowadays the community is not in a position to discern which point of view is the best, the authors should not present the work as if the theoretical framework is stacked in the 60's. A proper

introduction of the state of the art and the current important points of discussion would be more indicated to present their results.

Despite I acknowledge the important and careful experimental value of the work presented by the authors, I think that a proper discussion of the above questions is required in order for the work to be considered for publication.

Reviewer #2 (Remarks to the Author):

The paper presents a study of thermal conductivity in graphite ribbons with different levels of isotopic purity, which documents the effect of isotopic purity on the amplitude and temperature dependence of thermal conductivity and the hydrodynamic regime of Poiseuille flow. The authors conclude that there is “phonon Poiseuille flow in a 5 μm -wide suspended graphite ribbon with purified ^{13}C isotope concentration, while not in the case with natural abundance.”

For the reasons given below, I find the conclusion as formulated above unconvincing. On the other hand, the paper is a useful contribution to the ongoing quest for phonon hydrodynamics. A modified version of the paper with a weaker central claim may be suitable for publication in Nature Communications.

Here is a list of the issues to be addressed:

1. Figure 2 presents the raw experimental data. One can see that in the 80 K-100K temperature window, the thermal conductivity of isotopically purified graphite ribbon is roughly 20 percent larger than the natural graphite ribbon. This increase does point to a longer phonon mean-free-path in the isotopically-purified sample, but can hardly be considered as evidence for a qualitative difference (i.e. absence and presence of Poiseuille flow).
2. The authors do not compare their case with the consequences of isotopic purification in other materials like silicon or diamond.
3. They do not inform their readers that the maximum thermal conductivity of their isotopically-purified sample (1000 W/K.m) is lower than the thermal conductivity of bulk Highly-Oriented Bulk Graphite (HOPG), kish graphite or natural graphite.
4. Ref. 13 and 14 reported the observation of second sound in graphite samples with no isotope purification. If isotopic purification is necessary for Poiseuille flow, why is it not for second sound?
5. The main evidence for the presence of phonon Poiseuille flow is isotopically purified samples, and its absence in the natural ones, is given in Fig. 4. What is clear in the figure, is the presence of a shoulder in the cleaner sample which becomes softer in the dirtier one. The peak expected in calculations is absent in the experimental data. I do not see in the data any convincing evidence for the presence or the absence of the Poiseuille flow.
6. The paper has no discussion of the mean-free-path of phonons using experimentally measured specific heat and sound velocity.

In conclusion, the data in this paper implies that isotopic purification enhances the thermal conductivity of graphite by about twenty percent around 100 K. The data does not provide strong evidence for presence or absence of Poiseuille flow. However, admitting the presence of Poiseuille flow in this temperature range, the results do imply that it is deteriorated by isotopic impurities. The main claim of the paper needs to be tempered down.

Reviewer #3 (Remarks to the Author):

This work's primary contribution to the existing literature demonstrating evidence for phonon Poiseuille flow in graphite is its elegant use of time-domain thermoreflectance methods and micromachining of smaller-scale graphite flakes for measurement. In addition to examining a series of specimens with different widths (0.5-5 micrometers) and observing an increased tendency toward Poiseuille flow, they demonstrate that increased isotopic purity enhances Poiseuille flow. They also introduce a new criterion for inferring, from the thermal conductivity T-dependence, the presence of Poiseuille flow (by using the computed ballistic conductance for normalizing the data).

Overall I find the presentation to be well written and the experiments and data analysis to be sound. In terms of significance, I think it is fair to say the present results advance the knowledge on this topic modestly.

Overall these are nice results that are certainly worthy of publication. Something that I found missing from the narrative is a brief discussion (e.g. in the Results section) about size effects that suppress the thermal conductivity of the present very narrow samples from that of more bulk samples (e.g. those of Ref. 32). This is important since readers may be confused by the fact that the K values in the present work are about 4 times smaller at 100K than those of Ref. 32. Are the smaller K values consistent with the authors' theoretical work (Ref. 31)? Perhaps an additional graph (for an inset or for the Supplementary Info) showing K vs width at 100K for the present samples (including a data point for bulk HOPG and a theoretical curve) could be helpful here.

Reviewer #4 (Remarks to the Author):

In this work, the authors combine theoretical calculation and experimental measurement to investigate the phonon Poiseuille flow in isotopically-purified graphite ribbon. In the reviewer's best understanding, the authors claimed the observation of phonon Poiseuille flow via the super-ballistic temperature dependence of the thermal conductivity. However, two major concerns need to be addressed for the current manuscript to be considered for publication in nature communications.

Q1. This manuscript claims the super-ballistic temperature dependence thermal conductivity as a more rigorous criteria to confirm the phonon Poiseuille flow. However, to the reviewer's best knowledge, the criteria are also not rigorous. In the reviewer's understanding, boundary scattering in the phonon Poiseuille flow shows distinct temperature and size dependence. Based on the kinetic theory, the thermal conductivity could be written as:

$$k = \sum Cq v^2 \tau$$

Here the summation is over all the mode. Under phonon Poiseuille flow, the lifetime τ due to boundary scattering:

$$\tau \propto T^1 d^2$$

The mode wise average of is $\langle v^2 \tau \rangle = k / \langle Cq \rangle$ increasing with T is an indication of τ , which is reason to use ratio of $k / \langle Cq \rangle$ as a criterial for phonon Poiseuille flow

Similarly, the mode wise average of $\langle v \tau \rangle = k / \langle Cq v \rangle$ increase with T, will corresponds to the criterial in this manuscript i.e. k / G_{ball}

People can still use the mode wise average of $\langle \tau \rangle = k / \langle Cq v^2 \rangle$ as a criterial.

But all those criterial is not rigorous as even under the diffusive regime, all those three criterial could be met in materials with different spectrum. So all the criterial is necessary condition but not sufficient condition.

Q2. The rigorous criteria for the phonon Poiseuille flow is the super-linear size dependence. The reviewer is wondering whether the authors have investigated the size effect since sample of different widths are prepared.

Response letter for Observation of phonon Poiseuille flow in isotopically-purified graphite ribbons

Xin Huang^{1,†}, Yangyu Guo^{1,†}, Yunhui Wu¹, Satoru Masubuchi¹, Kenji Watanabe², Takashi Taniguchi^{1,3}, Zhongwei Zhang¹, Sebastian Volz^{1,4}, Tomoki Machida¹, and Masahiro Nomura^{1,5,*}

¹Institute of Industrial Science, The University of Tokyo, Tokyo 153-8505, Japan

²Research Center for Functional Materials, National Institute for Materials Science, Tsukuba 305-0044, Japan

³International Center for Materials Nanoarchitectonics, National Institute for Materials Science, Tsukuba 305-0044, Japan

⁴LIMMS, CNRS-IIS UMI 2820, The University of Tokyo, Tokyo 153-8505, Japan

⁵Research Center for Advanced Science and Technology, The University of Tokyo, Tokyo 153-0041, Japan

*corresponding author: Masahiro Nomura (nomura@iis.u-tokyo.ac.jp)

†these authors contributed equally to this work

Reviewer #1 (Remarks to the Author):

See also attachment.

The authors claim that they observe a non-monotonous behavior of the effective thermal conductivity of a graphite sample in terms of temperature. They claim that this is evidence of the emergence of a Poiseuille flow.

In the last years phonon hydrodynamics has been proposed to understand the non-Fourier behavior of thermal transport at the nanoscale. This has been very fruitful to understand 2D samples like graphene or samples like graphite that can be understood as a van der Waals stacking of graphene layers. In the last years there has been very intense efforts to experimentally observe it. Poiseuille flow is a phenomenon directly related with the hydrodynamic behavior of heat, and its direct observation has been pursued as evidence in favour of phonon hydrodynamics. These kinds of results are very interesting to the research community as they can help to identify in which situations one can expect hydrodynamics in thermal transport.

Despite the clear importance of the topic, and in particular the experiment that the authors present in their work, I have some criticism that to my opinion should be raised before publishing the work.

We appreciate the reviewer for the recognition of the importance of our work and the very pertinent comments for us to clarify the evidence of phonon Poiseuille flow in graphite ribbons. We have carefully considered the reviewer's comments or suggestions, and improved our manuscript correspondingly, as explained below.

a) The evidence in favour of the Poiseuille flow is based in the plots 4d/4e/4f. The authors base their arguments by claiming that there is a clear difference between plot 4f and plot 4d. From their point of view, there is a difference in the slope in the pure sample in 4f respect to that of 4d. Overlapping figure 4d with figures 4e-4f (see image in the attached pdf), it seems that there is no clear evidence in favour of the authors claims:

1) The observation of the peak near 100K it is not evident outside the statistical fluctuation in the measurements. The slope of the pure sample in 4f is not statistically different from the slope in figure 4d, where the authors claim that there is no Poiseuille flow. Even accepting that there is a peak near 60K, there is no explanation of the difference

with the peak observed near 100K in the same sample. The authors should explain the difference between both and give an statistical differentiation between them.

We provide a clearer statistical differentiation of the temperature-dependence of $\kappa/G_{\text{ballistic}}$ for the isotopically-purified graphite ribbons with different widths (from 500 nm to 5 μm) within the hydrodynamic temperature window, as shown in Fig. R1. As it is seen, with the rise of temperature, the values of $\kappa/G_{\text{ballistic}}$ monotonically decrease for 500 nm- and 1 μm -wide ribbons, showing no hydrodynamic behaviors due to the dominant phonon-boundary scattering in narrower structures. In the 3 μm -wide ribbon, the tendency of $\kappa/G_{\text{ballistic}}$ is flattened from 30 K and is attributed to the gradual enhancement of normal scattering in a wider structure. While the value of $\kappa/G_{\text{ballistic}}$ is enhanced by 16% from 30 to 50 K in the ribbon with a width of 5 μm , which clearly indicates the emergence of hydrodynamic phonon transport due to the dominance of momentum-conserved normal scattering.

Figure R1. The criterion and evidence of phonon Poiseuille flow in isotopically-purified graphite ribbons, namely, the ratio of thermal conductivity (κ) over ballistic thermal conductance ($G_{\text{ballistic}}$) as a function of temperature from 30 to 50 K within the hydrodynamic window. Solid lines show the linear fitting of experimental data.

2) The difference between samples with pure and natural isotope concentrations are almost identical for all the samples (4d, 4e, 4f). In the experimental data there is no evidence in favour of the differences predicted by the theoretical model used to interpret the data. In the solid lines (theoretical predictions) there is a clear tendency of the pure sample to merge with the natural sample. Once again, by overlapping plots 4c and 4e-4f this tendency is not clearly observed in the experimental data.

To provide a clearer demonstration of the difference between samples with pure and natural isotope concentrations in Figs. 4d-f, we compare the experimental results of both isotopically-purified and natural ribbons with the widths of 1 μm , 3 μm and 5 μm from 30 to 60 K (within the hydrodynamic window), as shown in Fig. R2. For narrower graphite ribbons, namely, 1 μm -wide ones, where normal scattering is weaker than the boundary scattering, both purified and natural samples show decreasing slopes as temperature increases. For 3 μm -wide ribbons, the natural sample still shows decreasing slope, while the slope of purified sample is flattened which is attributed to the enhancement of normal scattering. The different slopes come from the fact that the widening of the structure introduces more space for isotope-phonon scattering in the natural sample. Such a difference becomes more significant for 5 μm -wide ribbons where the hydrodynamic effect is stronger in the isotopically-purified sample, as shown in Fig. R2(c). In summary, our experimental data clearly differentiate the difference between samples with pure and natural isotope concentrations.

Figure R2. Thermal conductivity over ballistic thermal conductance ($\kappa/G_{\text{ballistic}}$) normalized by its value at 10 K as a function of temperature from 30 to 60 K for the graphite ribbons with the width of (a) 1 μm , (b) 3 μm and (c) 5 μm . The black (gray), red (pink), and dark blue (light blue) dots represent the experimental data of isotopically-purified (natural) graphite ribbons. Solid lines show the linear fitting of experimental data.

To answer the reviewer's concerns in question a), we added Fig. R2 in Supplementary as Supplementary Fig. 6 and the following text in the manuscript (page 6, lines 164-169). Note that the information of Fig. R1 is included in Fig. R2.

"Within the temperature window of phonon Poiseuille flow (i.e., 30–60 K), the slope of $\kappa/G_{\text{ballistic}}$ for isotopically-purified graphite ribbons evolves from a decreasing (1 μm) to a flattening (3 μm), and eventually an increasing (5 μm) trend with the rise of temperature, indicating a clear transition from ballistic regime to hydrodynamic regime similar to the observation in a recent theoretical work¹⁷. While the slope of $\kappa/G_{\text{ballistic}}$ for the natural counterparts oppositely drops faster with the widening of the ribbon (as detailed in Supplementary Fig. 6)."

b) Regarding the theoretical framework presented in the paper to identify the appearance of the Poiseuille flow, I have some concerns that to my opinion the authors should discuss.

In the first papers where Poiseuille flow has been observed, the behavior of thermal conductivity is compared to a T^3 curve. If the slope of thermal conductivity is larger than 3 in a log-log plot, Poiseuille flow is inferred. The theoretical explanation for this is clear in an isotropic material. At low temperatures, the mean free path saturates to the width of the sample. The velocity of the phonons is constant because in the low temperature region, the slope in the distribution function does not change. This leaves the specific heat as the only temperature dependent magnitude. This means that the test of the Poiseuille flow can be done in a pure experimental way by comparing the experimental data of thermal conductivity with that of the specific heat. As the slope of specific heat is constant, and only depending on dimensionality for 3D isotropic materials, this test offers no debate.

In 2D materials, this is more subtle. The quadratic band in the dispersion relations has an effect on the specific heat slope and the relation with the dimensionality is broken. This means that without access to experimental data on specific heat, its slope may depend on the theoretical model. The authors suggest that ballistic conductance should be done in order to solve this. That is, a temperature slope larger than the ballistic conductance should be used to identify Poiseuille flow.

Even accepting that the dimensionality cannot be used, the authors should present their results with a brief discussion of this important point. They should study the influence of the variation of the dispersion relations on the theoretical curves of the ballistic behaviour and the impact that may have on the conclusions of the paper.

We thank the reviewer for the pertinent comments and suggestion. Essentially the super-ballistic dependence of thermal conductivity in between the ballistic regime and the diffusive regime indicates the occurrence of hydrodynamic phonon transport. In most isotropic 3D materials, the ballistic limit of thermal conductance has the same temperature dependence as that of the heat capacity, i.e. following the classical Debye's T^3 law at extremely-low temperature. In contrast, for graphite with the bending acoustic phonon branch, the temperature scaling of heat capacity and ballistic thermal conductance are very different, as shown in Fig. R3. In principle, the results would depend on the used atomic interaction potential for theoretical calculations. As a comparison, we also calculate the heat capacity and ballistic thermal conductance based on an empirical atomic interaction potential [Phys. Rev. B 83, 235428 (2011)]. As is seen in Fig. R3(b), although there is some difference between the absolute values at low temperature, the temperature scaling of the ballistic thermal conductance are very close between the first-principles (DFT) calculation and the empirical potential. Therefore, the theoretical model shall have minor influence on the conclusion of our work.

To answer the reviewer's question b), we add some discussions in the main text (Page 5, lines 130-133) and in the Supplementary (Note 3 and Fig. 9) respectively:

"Note that we also calculate the ballistic thermal conductance based on an empirical atomic interaction potential, which has minor influence on its temperature dependence at low temperature and on the conclusion in the present work. More detailed discussions are given in Supplementary Note 3 and Supplementary Fig. 9."

"The calculation of the ballistic thermal conductance relies on the used atomic interaction potential. For a comparison, we also calculate the temperature-dependent heat capacity and ballistic thermal conductance of graphite based on an empirical potential⁶⁶. The optimized Tersoff potential and Lennard-Jones potential are adopted for the in-plane and inter-layer interactions respectively. As shown in Fig. S9(b), although there is some difference between the absolute values at low temperature, the temperature scaling behaviors of ballistic thermal conductance are almost the same by the first-principles (DFT) calculation and by the empirical potential. Therefore, the theoretical model shall have minor influence on the conclusion of our work. Still we recommend to adopt the first-principles method to calculate the ballistic thermal conductance as the obtained phonon dispersion better reproduces the experimental data of heat capacity, as seen in Fig. S9(a)."

Figure R3. Temperature scalings of (a) heat capacity and (b) ballistic thermal conductance of graphite. The discrete symbols represent the experimental data, whereas the solid line and dash-dotted line represent the calculation results by DFT (density functional theory) and by empirical atomic interaction potential respectively. The dashed line denotes the low-temperature limit of $C_p \sim T^{2.5}$ and $G_{ballistic} \sim T^{2.5}$. Both theoretical calculations are down to only 10 K due to insufficient resolution of the first Brillouin zone around Γ point below 10 K.

c) Regarding the novelty of the results, I should say that the observation Poiseuille flow in dielectrics is not new. It was experimentally observed on LiF and NaF in the 70's (<https://doi.org/10.1002/pssb.2220520127>) and very recently, a very similar work has been published in Science Advances (<https://doi.org/10.1126/sciadv.aat3374>) for Black Phosphorous. In this last work, the comparison with the experimental data of specific heat slope is done directly, and not through a ballistic conductance line.

Indeed the observation of phonon Poiseuille flow in solids is not totally new. It has been demonstrated in a limited number of materials since around 1970s at extremely low temperatures, for instance, He⁴ (< 0.9 K) [Sov. Phys. JETP 22, 47 (1966)], Bi (< 2.4 K) [Sov. J. Exp. Theor. Phys. 38, 357 (1974)], LiF and NaF (< 20K) [Phys. Status Solidi (B) 52, 253–262 (1972)], and recent work in black phosphorus (< 12 K) [Sci. Adv. 4, eaat3374 (2018)]. Nevertheless, in the present work, we demonstrate the evidence of phonon Poiseuille flow at record-high temperature (up to 60 K) attributed to the recent theoretical finding of strong hydrodynamic transport in graphitic materials, we also demonstrate the relevance of our novel understanding of the occurrence criterion. In this last work [Sci. Adv. 4, eaat3374 (2018)], the phonon dispersion of black phosphorous is fairly linear close to the Γ point. Thus both the thermal conductivity and the specific heat show a clear T^3 scaling at extremely low temperature. In other words, the present occurrence criterion through the ballistic thermal conductance is reduced to a criterion on the specific heat.

To clarify the importance and novelty of this work to the readers, we added the following text in the manuscript (page 9, lines 232-235).

"In light of a more relevant criterion due to the anisotropic nature of graphite, we observed prominent Poiseuille flow of phonons in an isotopically-purified graphite ribbon with a width of 5 μm up to 60 K, which is much elevated compared to the temperature range of previous observations in other materials^{29,51,52,59}."

d) In a more general scope, I should mention that in the scientific community there is a very important point of discussion regarding the microscopic justification for the appearance of the hydrodynamic behavior in thermal transport. The authors point of view follows the traditional point of view of the 60's initiated by Pierls and contemporary authors. They argue that the dominance of Normal collisions as the only justification for the

appearance of hydrodynamics. But in the last years, there has been a lot of works pointing to behaviours that are not in agreement with that point of view. Two important ones are the observation of second sound in Germanium (<https://doi.org/10.1126/sciadv.abg4677>), and (<https://doi.org/10.1038/s41467-021-27907-z>). The last one is in apparent contradiction with this work, as they observe second sound in natural isotopic graphite over 200K. From the point of view offered in the introduction of this work, these two observations cannot be explained. Despite nowadays the community is not in a position to discern which point of view is the best, the authors should not present the work as if the theoretical framework is stacked in the 60's. A proper introduction of the state of the art and the current important points of discussion would be more indicated to present their results.

We thank the reviewer's insightful comments for promoting hydrodynamic thermal transport in the scientific community of heat transfer. Indeed there are two different categories of second sound, i.e. 1) the "driftless" second sound, which does not require the dominance of normal scattering, exists when the energy flux decays slower than the characteristic time of experimental observation, as demonstrated in the very recent work in Ge with a rapidly varying temperature field [Sci. Adv. 7, eabg4677 (2021)]; 2) the traditional "drifting" one, which is justified by a commonly admitted condition that the excitation pulse frequency is smaller than the normal scattering rate but larger than the resistive scattering rate [Phys. Rev. 148, 778 (1966)].

The observation of the "drifting" second sound has been recently reported in a highly oriented pyrolytic graphite (HOPG) sample with natural isotope at 100 K [Science 364, 375–379 (2019)] followed by a very recent update at 200 K using an improved version of the same transient thermal grating (TTG) technique [Nat. Commun. 13, 1–9 (2022)]. Below we will explain why second sound could be observed in natural graphite while it is not the case for the phonon Poiseuille flow as explained in the present work.

Based on the hydrodynamic window condition ($\tau_N^{-1} \gg \Omega \gg \tau_R^{-1}$) [Phys. Rev. 148, 778 (1966)], the observation of second sound via TTG technique should satisfy: $l_N \ll l_g \ll l_R$, where l_g is the grating period of TTG. However, the window condition of the phonon Poiseuille flow is stricter: $l_N \ll W$, $l_R l_N \gg W^2$ [Phys. Rev. 148, 778 (1966)], where W is the ribbon width. In other words, l_R should be two and three orders of magnitudes larger than l_N to observe second sound and phonon Poiseuille flow, respectively. To have a more quantitative understanding, in Fig. R4, we show the MFPs of normal and resistive scatterings of the dominant bending acoustic (BA) phonons (at $k_z = 0$) in graphite at 60 K obtained by our first-principles modeling. Note that the resistive scattering here is basically presented by the isotope process since the Umklapp process is rare at lower temperatures. In both purified (0.02% ^{13}C) and natural (1.1% ^{13}C) cases, the sample width of 5 μm is much larger than the MFP of normal scattering ($l_N \ll W$). However, the MFP of isotope scattering in the natural ribbon is around one order of magnitude larger than the sample width (5 μm), such that $l_R l_N \sim W^2$. In other words, the graphite ribbon with a natural abundance of ^{13}C does not satisfy the window condition, which is instead valid in the purified ribbon since the MFP of isotope scattering is around two orders of magnitude larger than the sample width ($l_R l_N \gg W^2$). This explains why the phonon Poiseuille flow is only observed in the isotopically-purified graphite ribbon with a width of 5 μm while not in the natural graphite ribbon with the same width. For the case of the second sound, its evident observation in the recent report [Science 364, 375–379 (2019)] via TTG technique with a grating period $l_g \sim 10 \mu\text{m}$ at around 80 K is consistent with the window condition even in natural graphite, i.e. $l_N \ll l_g \ll l_R$, as inferred from Fig. R4.

Figure R4. Mean free paths of different phonon scattering processes in graphite at 60 K. The red and black circles denote the mean free path of normal and Umklapp scatterings respectively. The green and blue squares denote the mean free path of isotope scattering in natural (1.1% ^{13}C) and isotopically-purified (0.02% ^{13}C) graphite respectively. The results of the bending acoustic (BA) phonons (at $k_z = 0$) which dominate the hydrodynamic transport are shown here. The reference size of $5\ \mu\text{m}$ is the sample width of the present isotopically-purified and natural graphite ribbons.

To answer the reviewer's question d), we added the following text in the manuscript (page 1, lines 31-34) to provide more information about the past and current experimental observations of second sound:

"The drifting second sound was first observed unambiguously in solid He^4 crystals²¹, later in various other crystals²²⁻²⁷ with heat-pulse and light-scattering methods at low temperatures, whereas the driftless counterpart has been detected very recently in Ge even at room temperature under a rapidly varying temperature field¹⁹."

In addition, to provide more information about the window condition of phonon Poiseuille flow and second sound, and the isotope effect on their observation, we added Fig. R4 and the following text in the manuscript (page 8, Fig. 5; page 2, lines 41-43; page 8, lines 198-202) and discussion in Supplementary Note 4 respectively:

"Despite the numerous theoretical investigations of phonon Poiseuille flow in graphitic materials^{17,30,32,33}, the experimental observation remains challenging owing to its more stringent observation window condition compared to that of the second sound, as to be elucidated in this work."

"It explicitly demonstrates more stringent condition for the observation of phonon Poiseuille flow than that of second sound, which has been observed instead in natural graphite recently^{14,15,31}. We provide an explanation of the underlying reason by comparing the window conditions for phonon Poiseuille flow and second sound in Supplementary Note 4."

"A commonly admitted condition of the second sound is that the excitation pulse frequency is smaller than the normal scattering rate but larger than the resistive scattering rate ($\tau_N^{-1} \gg \Omega \gg \tau_R^{-1}$)¹³, or is equivalently the dominance of normal scattering over the resistive scattering ($l_N \ll l_{ex} \ll l_R$)⁵⁴, with l_{ex} referring to the length of the external excitation. The observation of second sound has been recently reported in the HOPG sample with natural isotope at 100 K¹⁴, followed by a very recent update at 200 K³¹ using an improved version of the same technique. In these two studies, the transient thermal grating (TTG) method was used to generate a periodically oscillating temperature field and the decay of the temperature

amplitude was measured to indicate the second sound. However, observing the steady-state hydrodynamic phenomenon, namely, the phonon Poiseuille flow, is expected to be more challenging than in the second sound case. As proposed by Guyer et al., phonon Poiseuille flow appears only under the following conditions⁵⁴: $l_N \ll W$, $l_R l_N \gg W^2$. Again, we adopt Fig. 5 in the main text for a more quantitative understanding of the isotope effect on observing phonon Poiseuille flow and second sound. As demonstrated and explained in the main text, the phonon Poiseuille flow is only observed in the isotopically-purified graphite ribbon with a width of 5 μm while not in the natural graphite ribbon with the same width. For the case of the second sound, the MFP of isotope scattering in the isotopically-purified sample and natural one is around three and two orders of magnitude larger than the MFP of normal scattering respectively, as seen in Fig. 5. Hence, the window condition to observe the second sound via TTG with a grating period in-between the MFPs of isotope scattering and normal scattering is satisfied in both purified and natural cases."

Despite I acknowledge the important and careful experimental value of the work presented by the authors, I think that a proper discussion of the above questions is required in order for the work to be considered for publication.

We thank the reviewer again for all the comments and suggestions. We hope our point-by-point response and additional discussions added to the manuscript may answer the reviewer's concerns on this work.

Reviewer #2 (Remarks to the Author):

The paper presents a study of thermal conductivity in graphite ribbons with different levels of isotopic purity, which documents the effect of isotopic purity on the amplitude and temperature dependence of thermal conductivity and the hydrodynamic regime of Poiseuille flow. The authors conclude that there is “phonon Poiseuille flow in a 5 μm -wide suspended graphite ribbon with purified ^{13}C isotope concentration, while not in the case with natural abundance.”

For the reasons given below, I find the conclusion as formulated above unconvincing. On the other hand, the paper is a useful contribution to the ongoing quest for phonon hydrodynamics. A modified version of the paper with a weaker central claim may be suitable for publication in Nature Communications.

We appreciate the reviewer for the recognition of the importance of our work and the very pertinent comments for us to clarify the evidence of phonon Poiseuille flow in graphite ribbons. We have carefully considered the reviewer’s comments or suggestions, and improved our manuscript correspondingly, as explained below.

Here is a list of the issues to be addressed:

1. Figure 2 presents the raw experimental data. One can see that in the 80K-100K temperature window, the thermal conductivity of isotopically purified graphite ribbon is roughly 20 percent larger than the natural graphite ribbon. This increase does point to a longer phonon mean-free-path in the isotopically-purified sample, but can hardly be considered as evidence for a qualitative difference (i.e. absence and presence of Poiseuille flow).

We thank the reviewer for the comments. We agree that the larger thermal conductivity in the isotopically-purified sample is associated with the longer apparent mean-free-path (MFP) of phonons compared to that of the natural sample. However, the enhancement of phonon MFP in the isotopically-purified sample facilitates the fulfillment of the hydrodynamic window condition and decisively relates to the presence of phonon Poiseuille flow, as explained below.

As raised by Guyer *et al.*, phonon Poiseuille flow appears in crystals under the following condition: $l_N \ll W$, $l_R l_N \gg W^2$ [Phys. Rev. 148, 778 (1966)], with l_N as the MFP of the normal processes, W as the ribbon width and l_R as the MFP of resistive processes. As depicted in Fig. R5, we show the MFPs of normal and resistive scatterings of the dominant bending acoustic (BA) phonons (at $k_z = 0$) in graphite at 60 K obtained by our first-principles modeling. In both purified (0.02% ^{13}C) and natural (1.1% ^{13}C) cases, the sample width of 5 μm is much larger than the MFP of normal scattering ($l_N \ll W$). However, the MFP of isotope scattering in the natural ribbon is around one order of magnitude larger than the sample width (5 μm), such that $l_R l_N \sim W^2$. In other words, the graphite ribbon with a natural abundance of ^{13}C does not satisfy the window condition, which is instead valid in the purified ribbon since the MFP of isotope scattering is around two orders of magnitude larger than the sample width ($l_R l_N \gg W^2$). Thus, the longer MFP of isotope (0.02% ^{13}C) scattering relevantly explains the presence of phonon Poiseuille flow in the 5 μm -wide isotopically-purified ribbon, while the one-order-magnitude shorter MFP of isotope (1.1% ^{13}C) scattering destroys phonon Poiseuille flow in the natural ribbon with the same width.

Figure R5. Mean free paths of different phonon scattering processes in graphite at 60 K. The red and black circles denote the mean free path of normal and Umklapp scatterings respectively. The green and blue squares denote the mean free path of isotope scattering in natural (1.1% ^{13}C) and isotopically-purified (0.02% ^{13}C) graphite respectively. The results of the bending acoustic (BA) phonons (at $k_z = 0$) which dominate the hydrodynamic transport are shown here. The reference size of $5\ \mu\text{m}$ is the sample width of the present isotopically-purified and natural graphite ribbons.

To answer the reviewer's concerns in question 1, we added the following text in the manuscript (page 3, lines 86-89) to highlight this important point:

"The isotope scattering plays an important role in this regime, resulting in the enhanced thermal conductivity of isotopically-purified graphite ribbon compared to that of the natural one by 21% at 100 K, which also vitally affects the occurrence of phonon Poiseuille flow as discussed in the following content."

Furthermore, we added Fig. R5 and the following discussion in the manuscript (page 8, Fig. 5; page 7-8, lines 188-198):

"To have a more quantitative understanding, in Fig. 5, we show the MFPs of normal and resistive scatterings of bending acoustic (BA) phonons (at $k_z = 0$) in natural and isotopically-purified graphite at 60 K obtained by our first-principles modeling. Note that the resistive scattering here is basically presented by the isotope process since the Umklapp process is rare at lower temperatures. In both purified (0.02% ^{13}C) and natural (1.1% ^{13}C) cases, the sample width of $5\ \mu\text{m}$ is much larger than the MFP of normal scattering ($l_N \ll W$). However, the MFP of isotope scattering in the natural ribbon is around one order of magnitude larger than the sample width ($5\ \mu\text{m}$), such that $l_R l_N \sim W^2$. In other words, the graphite ribbon with a natural abundance of ^{13}C does not satisfy the window condition, which is instead valid in the purified ribbon since the MFP of isotope scattering is around two orders of magnitude larger than the sample width ($l_R l_N \gg W^2$). This explains why the phonon Poiseuille flow is only observed in the isotopically-purified graphite ribbon with a width of $5\ \mu\text{m}$ while not in the natural graphite ribbon with the same width."

2. The authors do not compare their case with the consequences of isotopic purification in other materials like silicon or diamond.

We thank the reviewer for this relevant suggestion. To compare the effect of isotopic purification in graphite and other materials, we first added the following text in the manuscript (page 3, lines 92-95):

"The effect of isotopic enrichment on the increase of thermal conductivity has been also observed in many other materials within the temperature range of 128–380 K, such as Si (10%)⁴⁴, GaN (15%)⁴⁵, boron phosphide (17%)⁴⁶, graphene (36%)⁴⁷, and diamond (50%)⁴⁸."

In addition, we also examined the isotope effect on the observation of phonon Poiseuille flow based on the present criterion ($\kappa/G_{\text{ballistic}}$) in silicon samples with both natural and enriched ^{28}Si isotope concentrations. As seen in Fig. R6, the isotopic purification leads to around 7.5 times higher thermal conductivity than the natural silicon sample at 26.5 K [Phys. Status Solidi (C) 1, 2995–2998 (2004)]. However, $\kappa/G_{\text{ballistic}}$ of both natural and purified silicon samples monotonically drops as temperature increases. In other words, phonon Poiseuille flow is not present even in the isotopically-purified silicon sample due to the insufficient normal scattering to satisfy the window condition. The absence of phonon Poiseuille flow in silicon also agrees well with previous demonstrations [Phys. Rev. Lett. 120, 125901 (2018); Sci. Adv. 4, eaat3374 (2018)]. For the case of diamond, despite the high Debye temperature and stronger normal scattering than the resistive one, the window between normal and resistive scatterings is extremely narrow to showcase phonon Poiseuille flow even in an isotopically-purified (0.01% ^{13}C) sample owing to the weak anharmonicity of diamond, while the window is fully closed in the diamond sample with 0.1% ^{13}C , as predicted by a previous work [Nat. Commun. 6, 1–10 (2015)].

Figure R6. Examining the criterion of phonon Poiseuille flow in silicon. The ratio of thermal conductivity (κ) over ballistic thermal conductance ($G_{\text{ballistic}}$) (the present criterion) as a function of temperature for bulk silicon with purified (99.983%) and natural abundance (92.23%) ^{28}Si isotope. The experimental thermal conductivity of silicon is obtained from [Phys. Status Solidi (C) 1, 2995–2998 (2004)]. The ballistic thermal conductance for silicon is calculated by the first-principles method.

Following the reviewer's suggestions in comment 2, we added Fig. R6 in Supplementary as Supplementary Fig. 7 and the following discussion in the manuscript (page 8, lines 203-207) to compare the isotope effects on the observation of phonon Poiseuille flow in common 3D materials like silicon, and graphite in the present work:

"In addition, we examined the isotope effect on the observation of phonon Poiseuille flow based on the present criterion ($\kappa/G_{\text{ballistic}}$) in silicon samples with natural and purified ^{28}Si isotope concentrations⁴⁴ (Supplementary Fig. 7). As temperature increases, $\kappa/G_{\text{ballistic}}$ of both natural and purified silicon samples drops monotonically, indicating the absence of phonon Poiseuille flow even in the isotopically-purified silicon sample. This is explained by the well-known insufficient normal scattering in silicon to satisfy the hydrodynamic window condition."

3. They do not inform their readers that the maximum thermal conductivity of their isotopically-purified sample (1000 W/K.m) is lower than the thermal conductivity of bulk Highly-Oriented Bulk Graphite (HOPG), kish graphite or natural graphite.

To provide more information to the readers, we compared the thermal conductivity of our isotopically-purified graphite ribbon with a width of 5 μm to the value of bulk HOPG (width: 350 μm) in a very recent report [Science 367, 309–312 (2020)]. As seen in Fig. R7, the thermal conductivity results show a generally similar trend of temperature dependence and a peak around 150 K. Our measured maximum value is $\sim 1330 \text{ Wm}^{-1}\text{K}^{-1}$, which is lower than that of the bulk HOPG ($\sim 2300 \text{ Wm}^{-1}\text{K}^{-1}$). We conclude that the reduction of thermal conductivity in our submicroscale ribbons has resulted from the strong size effect. The similar size effect arising from the downscaling of the bulk graphitic sample was also demonstrated in previous experimental and theoretical works [Nat. Commun. 4, 1–7 (2013); Nat. Commun. 5, 1–6 (2014); Nano Lett. 14, 6109–6114 (2014)].

Following the reviewer's suggestion, we added Fig. R7 in Supplementary as Supplementary Fig. 4a and the following text in the manuscript (page 3, lines 82-84):

"The peak value of isotopically-purified sample is measured as $\sim 1330 \text{ Wm}^{-1}\text{K}^{-1}$, which is lower than that of bulk HOPG³⁴ reported in a recent work owing to the strong size effect from structure downscaling^{41–43} (Supplementary Fig. 4)."

Figure R7. Temperature-dependent in-plane thermal conductivity of 5 μm -wide isotopically-purified graphite ribbon, compared with the value of bulk HOPG [Science 367, 309–312 (2020)].

4. Ref. 13 and 14 reported the observation of second sound in graphite samples with no isotope purification. If isotopic purification is necessary for Poiseuille flow, why is it not for second sound?

This is indeed a pertinent question. As mentioned by the reviewer, the observation of second sound has been recently reported in a highly oriented pyrolytic graphite (HOPG) sample with natural isotope at 100 K [Science 364, 375–379 (2019)] followed by a very recent update at 200 K using an improved version of the same transient thermal grating (TTG) technique [Nat. Commun. 13, 1–9 (2022)]. Below we will explain why the isotopic purification is necessary for the observation of phonon Poiseuille flow in graphite while not for the case of second sound.

Based on the hydrodynamic window condition ($\tau_N^{-1} \gg \Omega \gg \tau_R^{-1}$) [Phys. Rev. 148, 778 (1966)], the observation of second sound via TTG technique should satisfy: $l_N \ll l_g \ll l_R$, where l_g is the grating period of TTG. However, the window condition of the phonon Poiseuille flow is stricter: $l_N \ll W$, $l_R l_N \gg W^2$ [Phys. Rev. 148, 778 (1966)], where

W is the ribbon width. In other words, l_R should be two and three orders of magnitudes larger than l_N to observe second sound and phonon Poiseuille flow, respectively. To have a more quantitative understanding, in Fig. R8, we show the MFPs of normal and resistive scatterings of the dominant bending acoustic (BA) phonons (at $k_z = 0$) in graphite at 60 K obtained by our first-principles modeling. Note that the resistive scattering here is basically presented by the isotope process since the Umklapp process is rare at lower temperatures. In both purified (0.02% ^{13}C) and natural (1.1% ^{13}C) cases, the sample width of 5 μm is much larger than the MFP of normal scattering ($l_N \ll W$). However, the MFP of isotope scattering in the natural ribbon is around one order of magnitude larger than the sample width (5 μm), such that $l_R l_N \sim W^2$. In other words, the graphite ribbon with a natural abundance of ^{13}C does not satisfy the window condition, which is instead valid in the purified ribbon since the MFP of isotope scattering is around two orders of magnitude larger than the sample width ($l_R l_N \gg W^2$). This explains why the phonon Poiseuille flow is only observed in the isotopically-purified graphite ribbon with a width of 5 μm while not in the natural graphite ribbon with the same width. For the case of the second sound, its evident observation in the recent report [Science 364, 375–379 (2019)] via TTG technique with a grating period $l_g \sim 10 \mu\text{m}$ at around 80 K is consistent with the window condition even in natural graphite, i.e. $l_N \ll l_g \ll l_R$, as inferred from Fig. R8.

Figure R8. Mean free paths of different phonon scattering processes in graphite at 60 K. The red and black circles denote the mean free path of normal and Umklapp scatterings respectively. The green and blue squares denote the mean free path of isotope scattering in natural (1.1% ^{13}C) and isotopically-purified (0.02% ^{13}C) graphite respectively. The results of the bending acoustic (BA) phonons (at $k_z = 0$) which dominate the hydrodynamic transport are shown here. The reference size of 5 μm is the sample width of the present isotopically-purified and natural graphite ribbons.

To answer the reviewer's concerns in question 4 and compare the window conditions of phonon Poiseuille flow and second sound, and the isotope effect on their observation, we added Fig. R8 and the following text in the manuscript (page 8, Fig. 5; page 2, lines 41-43; page 8, lines 198-202) and discussion in Supplementary Note 4 respectively:

"Despite the numerous theoretical investigations of phonon Poiseuille flow in graphitic materials^{17,30,32,33}, the experimental observation remains challenging owing to its more stringent observation window condition compared to that of the second sound, as to be elucidated in this work."

"It explicitly demonstrates more stringent condition for the observation of phonon Poiseuille flow than that of second sound,

which has been observed instead in natural graphite recently^{14,15,31}. We provide an explanation of the underlying reason by comparing the window conditions for phonon Poiseuille flow and second sound in Supplementary Note 4."

"A commonly admitted condition of the second sound is that the excitation pulse frequency is smaller than the normal scattering rate but larger than the resistive scattering rate ($\tau_N^{-1} \gg \Omega \gg \tau_R^{-1}$)¹³, or is equivalently the dominance of normal scattering over the resistive scattering ($l_N \ll l_{ex} \ll l_R$)⁵⁴, with l_{ex} referring to the length of the external excitation. The observation of second sound has been recently reported in the HOPG sample with natural isotope at 100 K¹⁴, followed by a very recent update at 200 K³¹ using an improved version of the same technique. In these two studies, the transient thermal grating (TTG) method was used to generate a periodically oscillating temperature field and the decay of the temperature amplitude was measured to indicate the second sound. However, observing the steady-state hydrodynamic phenomenon, namely, the phonon Poiseuille flow, is expected to be more challenging than in the second sound case. As proposed by Guyer et al., phonon Poiseuille flow appears only under the following conditions⁵⁴: $l_N \ll W$, $l_R l_N \gg W^2$. Again, we adopt Fig. 5 in the main text for a more quantitative understanding of the isotope effect on observing phonon Poiseuille flow and second sound. As demonstrated and explained in the main text, the phonon Poiseuille flow is only observed in the isotopically-purified graphite ribbon with a width of 5 μm while not in the natural graphite ribbon with the same width. For the case of the second sound, the MFP of isotope scattering in the isotopically-purified sample and natural one is around three and two orders of magnitude larger than the MFP of normal scattering respectively, as seen in Fig. 5. Hence, the window condition to observe the second sound via TTG with a grating period in-between the MFPs of isotope scattering and normal scattering is satisfied in both purified and natural cases."

5. The main evidence for the presence of phonon Poiseuille flow is isotopically purified samples, and its absence in the natural ones, is given in Fig. 4. What is clear in the figure, is the presence of a shoulder in the cleaner sample which becomes softer in the dirtier one. The peak expected in calculations is absent in the experimental data. I do not see in the data any convincing evidence for the presence or the absence of the Poiseuille flow.

We thank the reviewer for the comments for us to improve this crucial point. We try to make the evidence of phonon Poiseuille flow clearer and convincing as detailed below:

To provide a more explicit demonstration of the difference between samples with pure and natural isotope concentrations in Figs.4d-f, we compare the experimental results of both isotopically-purified and natural ribbons with the width of 1 μm , 3 μm and 5 μm from 30 to 60 K (within the hydrodynamic window), as shown in Fig. R9. Both purified and natural samples show decreasing slopes as temperature increases for narrower graphite ribbons, namely, the 1 μm -wide ones, where normal scattering is weaker than the boundary scattering. The decreasing slope of the natural one is slightly steeper than the purified one due to the isotope effect, i.e., the shoulder became softer as mentioned by the reviewer. However, for 3 μm -wide ribbons, the natural sample still shows decreasing slope but steeper than the one of the 1 μm -wide case. In contrast, the slope in the purified sample case is more or less flattened, which is attributed to the enhancement of normal scattering. The different slopes come from the fact that the widening of the structure introduces more space for isotope-phonon scattering in natural sample. Such a difference becomes more significant for 5 μm -wide ribbons where the hydrodynamic effect is more substantial in the isotopically-purified sample, as shown in Fig. R9(c).

In addition, to clarify the evidence for the presence of phonon Poiseuille flow, we provide a clearer statistical differentiation of the temperature-dependence of $\kappa/G_{\text{ballistic}}$ for the isotopically-purified graphite ribbons with different widths (from 500 nm to 5 μm) within the hydrodynamic temperature window, as shown in Fig. R10. As it is seen, with the raise of temperature, the values of $\kappa/G_{\text{ballistic}}$ monotonically decrease for 500 nm- and 1 μm -wide ribbons, showing no hydrodynamic behaviors due to the dominant phonon-boundary scattering in narrower structures. In the 3 μm -wide ribbon, the tendency of $\kappa/G_{\text{ballistic}}$ is flattened from 30 K, which is attributed to the gradual enhancement of normal scattering in a wider structure. While the value of $\kappa/G_{\text{ballistic}}$ is enhanced by 16% from 30 to 50 K in the ribbon with a width of 5 μm , which clearly indicates the emergence of hydrodynamic phonon transport due to the dominance of momentum-conserving normal scattering.

As temperature further increases beyond the hydrodynamic window (i.e., 30–70 K, as predicted by our theoretical calculation), $\kappa/G_{\text{ballistic}}$ would monotonically decrease due to the increasing rate of Umklapp scattering. For the case of 5 μm -wide isotopically-purified ribbon shown in Fig. 4f, the destruction of phonon Poiseuille flow occurs at the temperature over 60 K in our experimental result, which is earlier than the drop of $\kappa/G_{\text{ballistic}}$ at 70 K in our

calculated result. It might be caused by the unknown defect or contamination in sample fabrication which introduces additional resistive scattering and shortens the hydrodynamic temperature window to 30–60 K, or uncertainty in the thermal conductivity measurement beyond 60 K.

Figure R9. Thermal conductivity over ballistic thermal conductance ($\kappa / G_{\text{ballistic}}$) normalized by its value at 10 K as a function of temperature from 30 to 60 K for the graphite ribbons with the width of (a) 1 μm , (b) 3 μm and (c) 5 μm . The black (gray), red (pink), and dark blue (light blue) dots represent the experimental data of isotopically-purified (natural) graphite ribbons. Solid lines show the linear fitting of experimental data.

Figure R10. The criterion and evidence of phonon Poiseuille flow in isotopically-purified graphite ribbons, namely, the ratio of thermal conductivity (κ) over ballistic thermal conductance ($G_{\text{ballistic}}$) as a function of temperature from 30 to 50 K within the hydrodynamic window. Solid lines show the linear fitting of experimental data.

To answer the reviewer's concerns in question 5, we added Fig. R9 in Supplementary as Supplementary Fig. 6 and

the following text in the manuscript (page 6, lines 164-169). Note the information of Fig. R10 is included in Fig. R9.

"Within the temperature window of phonon Poiseuille flow (i.e., 30–60 K), the slope of $\kappa/G_{\text{ballistic}}$ for isotopically-purified graphite ribbons evolves from a decreasing ($1\ \mu\text{m}$) to a flattening ($3\ \mu\text{m}$), and eventually an increasing ($5\ \mu\text{m}$) trend with the rise of temperature, indicating a clear transition from ballistic regime to hydrodynamic regime similar to the observation in a recent theoretical work¹⁷. While the slope of $\kappa/G_{\text{ballistic}}$ for the natural counterparts oppositely drops faster with the widening of the ribbon (as detailed in Supplementary Fig. 6)."

Moreover, we added the following text in the manuscript (page 6, lines 169-175) to explain the different peak positions between experimental and calculated results:

"As temperature further increases beyond the hydrodynamic window, $\kappa/G_{\text{ballistic}}$ would decrease due to the increasing Umklapp scattering rate. For the case of $5\ \mu\text{m}$ -wide isotopically-purified ribbon shown in Fig. 4f, the destruction of phonon Poiseuille flow occurs at the temperature over 60 K in our experimental result, which is slightly earlier than the drop of $\kappa/G_{\text{ballistic}}$ at 70 K in our calculation. It might be caused by the unknown defect or contamination during sample fabrication which introduces additional resistive scattering and shortens the hydrodynamic temperature window to 30–60 K, or uncertainty in thermal conductivity measurement beyond 60 K."

6. The paper has no discussion of the mean-free-path of phonons using experimentally measured specific heat and sound velocity.

We thank the reviewer for the suggestion. Indeed, from the temperature-dependence of effective momentum-destroying MFP of phonon (l_{eff}), one could indicate the presence of phonon Poiseuille flow. In the hydrodynamic regime, $l_{\text{eff}} \sim W^2/l_N$ from the random walk theory [Sov. Phys. JETP 19, 490 (1964); Sov. Phys. Uspekhi 11, 255 (1968)], with l_N as the MFP of the normal process. Therefore, as the temperature increases, the enhancement of normal scattering (l_N decreases) results in the increase of l_{eff} and evidences the phonon Poiseuille flow. The increase of l_{eff} has been extracted based on the kinetic theory, $l_{\text{eff}} \sim \kappa/Cv$, in 3D materials close to isotropic structures with constant sound velocity, such as Bi [Sov. J. Exp. Theor. Phys. 38, 357 (1974)] and BP [Sci. Adv. 4, eaat3374 (2018)]. For graphite, the sound velocity in the long-wavelength limit will not be simply a constant due to the quadratic dispersion of the bending acoustic phonons. Nevertheless, the present criterion to evidence the phonon Poiseuille flow, i.e., $\kappa/G_{\text{ballistic}}$, is somehow equivalent to κ/Cv , or l_{eff} . Actually $\kappa/G_{\text{ballistic}}$ has the unit of a length, as shown in Fig. 3(b) in the main text.

Following the reviewer's suggestion. We added the following text in the manuscript (page 4, lines 105-107) to discuss this important point:

"One may feature the hydrodynamic thermal transport in principle from the temperature-dependence of l associated with the strength of the normal process, as detected in some crystals^{29,51}."

In conclusion, the data in this paper implies that isotopic purification enhances the thermal conductivity of graphite by about twenty percent around 100 K. The data does not provide strong evidence for presence or absence of Poiseuille flow. However, admitting the presence of Poiseuille flow in this temperature range, the results do imply that it is deteriorated by isotopic impurities. The main claim of the paper needs to be tempered down.

We thank the reviewer again for all the comments and suggestions. We hope our point-by-point response and additional discussions added to the manuscript may answer the reviewer's concerns on this work.

Reviewer #3 (Remarks to the Author):

This work's primary contribution to the existing literature demonstrating evidence for phonon Poiseuille flow in graphite is its elegant use of time-domain thermoreflectance methods and micromachining of smaller-scale graphite flakes for measurement. In addition to examining a series of specimens with different widths (0.5-5 micrometers) and observing an increased tendency toward Poiseuille flow, they demonstrate that increased isotopic purity enhances Poiseuille flow. They also introduce a new criterion for inferring, from the thermal conductivity T-dependence, the presence of Poiseuille flow (by using the computed ballistic conductance for normalizing the data).

Overall I find the presentation to be well written and the experiments and data analysis to be sound. In terms of significance, I think it is fair to say the present results advance the knowledge on this topic modestly.

Overall these are nice results that are certainly worthy of publication. Something that I found missing from the narrative is a brief discussion (e.g. in the Results section) about size effects that suppress the thermal conductivity of the present very narrow samples from that of more bulk samples (e.g. those of Ref. 32). This is important since readers may be confused by the fact that the K values in the present work are about 4 times smaller at 100K than those of Ref. 32. Are the smaller K values consistent with the authors' theoretical work (Ref. 31)? Perhaps an additional graph (for an inset or for the Supplementary Info) showing K vs width at 100K for the present samples (including a data point for bulk HOPG and a theoretical curve) could be helpful here.

We appreciate the reviewer for the recognition of the importance of our work and further suggestions to enrich the contents of our manuscript. We have carefully considered the reviewer's suggestion, and improved our manuscript correspondingly, as explained below.

Following the reviewer's suggestion, we compared the thermal conductivities of our submicroscale graphite ribbons with the widths from 500 nm to 5 μm at 150 K to the values of bulk HOPG (width: 350 μm) in a very recent work [Science 367, 309–312 (2020)] which span from $\sim 1200 \text{ Wm}^{-1}\text{K}^{-1}$ to $\sim 4500 \text{ Wm}^{-1}\text{K}^{-1}$ with varying thickness, as shown in Fig. R11. Our measured thermal conductivity is $\sim 1330 \text{ Wm}^{-1}\text{K}^{-1}$, which is lower than that of the bulk HOPG values in general. And we conclude that the reduction of thermal conductivity in our submicroscale-wide ribbons is resulting from the strong size effect. The similar size effect from downscaling of the bulk graphitic sample has been also demonstrated in other experimental and theoretical works [Nat. Commun. 4, 1–7 (2013); Nat. Commun. 5, 1–6 (2014); Nano Lett. 14, 6109–6114 (2014)]. The absolute value of our experimental data is lower than that of our modelling results by BTE with first-principles inputs, which might be caused by the additional resistive scattering of phonons induced by unknown defect or contamination in sample fabrication. However, our experimental width-dependence of thermal conductivity shows good qualitative consistency with our calculated results. Moreover, both experimental and calculated results of $\kappa/G_{\text{ballistic}}$ are normalized by their values at 10 K (ballistic limit), which show very consistent trends to demonstrate the hydrodynamic phonon transport in this work.

To provide more information to the readers, we added Fig. R11 in Supplementary as Supplementary Fig. 4b and the following text in the manuscript (page 3, lines 82-84).

"The peak value of isotopically-purified sample is measured as $\sim 1330 \text{ Wm}^{-1}\text{K}^{-1}$, which is lower than that of bulk HOPG³⁴ reported in a recent work owing to the strong size effect from structure downscaling^{41–43} (Supplementary Fig. 4)."

Figure R11. Width-dependent in-plane thermal conductivity of isotopically-purified graphite ribbon at 150 K, compared with the range of values of bulk HOPG with varying thickness [Science 367, 309–312 (2020)].

Reviewer #4 (Remarks to the Author):

Please see the attachment.

In this work, the authors combine theoretical calculation and experimental measurement to investigate the phonon Poiseuille flow in isotopically-purified graphite ribbon. In the reviewer's best understanding, the authors claimed the observation of phonon Poiseuille flow via the super-ballistic temperature dependence of the thermal conductivity. However, two major concerns need to be addressed for the current manuscript to be considered for publication in nature communications.

We appreciate the reviewer's time in reading and understanding our work and the very pertinent comments for us to clarify the rigor of the criterion for demonstrating phonon Poiseuille flow in graphite ribbons. We have carefully considered the reviewer's comments or suggestions, and improved our manuscript correspondingly, as explained below.

Q1. This manuscript claims the super-ballistic temperature dependence thermal conductivity as a more rigorous criteria to confirm the phonon Poiseuille flow. However, to the reviewer's best knowledge, the criteria are also not rigorous. In the reviewer's understanding, boundary scattering in the phonon Poiseuille flow shows distinct temperature and size dependence. Based on the kinetic theory, the thermal conductivity could be written as:

$$k = \sum Cqv^2\tau$$

Here the summation is over all the mode. Under phonon Poiseuille flow, the lifetime due to boundary scattering:

$$\tau \propto T^1 d^2$$

The mode wise average of $\langle v^2\tau \rangle = k / \langle Cq \rangle$ increasing with T is an indication of τ , which is reason to use ratio of $k / \langle Cq \rangle$ as a criterial for phonon Poiseuille flow

Similarly, the mode wise average of $\langle v\tau \rangle = k / \langle Cqv \rangle$ increase with T, will corresponds to the criterial in this manuscript i.e. k/G_{ball}

People can still use the mode wise average of $\langle \tau \rangle = k / \langle Cqv^2 \rangle$ as a criterial.

But all those criterial is not rigorous as even under the diffusive regime, all those three criterial could be met in materials with different spectrum. So all the criterial is necessary condition but not sufficient condition.

We agree with the reviewer's opinion. Indeed, all those criteria are not strictly rigorous due to the complex spectral dependence of phonon properties in crystals. However, we could only use the most rigorous one among them. As mentioned by the reviewer, thermal conductivity (κ) could be approximately expressed as: $\kappa \sim Cv l = Cv^2\tau$. In the hydrodynamic regime, the effective momentum-destroying MFP is obtained from the random walk theory as [Sov. Phys. JETP 19, 490 (1964); Sov. Phys. Uspekhi 11, 255 (1968)]: $l \sim d^2/l_N$, or equivalently $\tau \sim d^2/(l_N v)$, with l_N as the MFP of the normal process. In 3D materials close to isotropic structures, phonon velocity (v) is approximately the speed of sound as a constant due to the linear dispersion relation of acoustic phonons. Therefore, as the temperature increases, the enhancement of normal scattering (l_N decreases) results in the increase of l or τ , as the evidence of the phonon Poiseuille flow. In this situation, all the three aforementioned criteria (i.e. κ/C , κ/Cv and κ/Cv^2) are equivalent, as reported in He⁴ [Sov. Phys. JETP 22, 47 (1966)], Bi [Sov. J. Exp. Theor. Phys. 38, 357 (1974)], SrTiO₃ [Phys. Rev. Lett. 120, 125901 (2018)], and BP [Sci. Adv. 4, eaat3374 (2018)].

In contrast, the anisotropic nature of graphite makes the situation different. The group velocity of bending acoustic (BA) modes in graphite increases with frequency due to the quadratic dispersion curve [Nat. Commun. 6, 1–10 (2015)]. Therefore, as the temperature increases, both the enhancement of normal scattering (l_N decreases) and temperature-dependent average group velocity of BA phonons contribute to the variation of τ . Hence, to disentangle the contributions from normal process and v , the rise of l , or equivalently κ/Cv ($\kappa/G_{\text{ballistic}}$) is a more relevant criterion. As shown in Fig. 3(a) in the main text, $\kappa/T^{2.5}$ (i.e. κ/C , since $C \sim T^{2.5}$ in graphite) increases with temperature from 10 to ~ 50 K for all four isotopically-purified graphite ribbons with different widths, including

even the case of 500 nm-wide ribbon, where the heat transport should lie within the ballistic regime. This could be explained by the increase of v of BA modes in the same temperature range. In other words, the criterion κ/C (or $\kappa/T^{2.5}$) does not definitely indicate the occurrence of hydrodynamic phonon flow in graphite. Of course, people might even use the criterion κ/Cv^2 . However, we prefer to use κ/Cv ($\kappa/G_{\text{ballistic}}$) since Cv has a clearer physical interpretation (i.e. ballistic thermal conductance).

Q2. The rigorous criteria for the phonon Poiseuille flow is the super-linear size dependence. The reviewer is wondering whether the authors have investigated the size effect since sample of different widths are prepared.

We thank the reviewer for the nice suggestion to investigate the size dependence for evidencing the phonon Poiseuille flow in graphite ribbons.

Indeed the super-linear width dependence of thermal conductivity is a very strong evidence of the phonon Poiseuille flow, as indicated by the effective mean free path $l \sim W^2/l_N$. To this end, we plot our experimentally measured thermal conductivity as a function of ribbon width of the isotopically-purified sample at three typical temperatures, as seen in Fig. R12. We have not observed the super-linear size dependence of thermal conductivity. However, we could clearly see that the width-dependent exponent α within the hydrodynamic temperature window (50 K) is appreciably larger than that outside the window (20 K and 160 K). The super-linear size dependence represents even stronger hydrodynamic phonon flow and is more difficult to observe because of the additional size effect from the ribbon length, as shown in our very recent theoretical investigation [Phys. Rev. B 104, 075450(2021)]. A relevant experiment is still going on in our group to investigate this important point as proposed by the reviewer. In summary, the experimental evidence of the phonon Poiseuille flow from the temperature-dependence of $\kappa/G_{\text{ballistic}}$ (or l) is the main contribution in the present work. Additionally, we added the following text about the evidence of phonon Poiseuille flow from the other perspective of width-dependence in the manuscript (page 9, lines 224-230) as the outlook of this work.

Figure R12. In-plane thermal conductivity as a function of the width of isotopically-purified graphite ribbon at 20, 50 and 160 K. Solid lines show the linear fitting of experimental data.

"According to the different temperature-dependent behaviors of $\kappa/G_{\text{ballistic}}$ in our isotopically-purified graphite ribbons with various widths, we observe the transition from the ballistic to the hydrodynamic thermal transport when the ribbon width increases from 500 nm to 5 μm . Besides, another important aspect to evidence the phonon hydrodynamic flow is the super-ballistic width dependence of thermal conductivity, as clearly indicated by the effective mean free path $l \sim W^2/l_N$. The suspended microstructure system built up in this work provides a good platform for further experiments to investigate the

extraordinary super-linear width dependence of thermal conduction in the hydrodynamic regime or the phonon Knudsen minimum phenomenon^{17,32,33,58}."

We thank the reviewer again for all the comments and suggestions. We hope that our point-by-point response and additional discussions added to the manuscript may answer the reviewer's concerns on this work.

Reviewers' comments:

Reviewer #1 (Remarks to the Author):

The authors have not answered the questions and doubts regarding my previous report.

a) Regarding the questions about the statistical significance of the increasing in thermal conductivity respect the ballistic conductivity, the authors have limited to include a zoomed plot with the data from 30 to 50 K.

1. When I pointed to the **statistical** significance of the maximum, I was not indicating that I didn't see an increasing trend between that temperature ranges (that was obvious). My concern was about that there is not a clear work to show that this trend was not because of experimental error. I explain this point in order to clarify it.

Looking at the image on the left, the justification in favour of a Poiseuille flow is based in the fact that in region 1 there is a maximum in the normalized thermal conductivity. In order for this to be **statistically acceptable**, the authors should include some argumentation about the minimum at region 2 and a new local maximum in region 3. From my point of view, these are showing the large experimental error of the measure. Without the proper description of

these fluctuations, the argument that in region 1 we are observing the emergence of a new phenomenon is statistically loose.

I am aware that the authors are in the limit of resolution of a very challenging measure. In that case, they have an alternative to solve this problem. They can use a theoretical model where the predictions demonstrate the appearance of this emergent phenomena in the same place where the data seems to be indicating it. In that case, the presented model (based on the BTE) does not seem to be indicating this. In the following plot, the model (solid lines) is predicting an increase in the reduced thermal conductivity in both regions (1 and 2).

The authors suggestion that the increase of thermal conductivity in region 2 is a clear evidence of Poiseuille should be accompanied with a model where in region 1 is not predicting this increase.

2. In point 2, the authors use again the zoom to evidence the change in the trend of their data in the 30-50 K range. This does not respond my concern about the lack of predictability of the model they are using.

As it can be seen in the figure on the right, the black arrows and red arrows show a clear tendency in the model where the pure and natural abundance samples split when increasing its size. This is clearly not observed

in the experimental data. In my previous report I combined the plots to show this. In this figure it can be observed that the red and black dots (data in the ellipse) are at similar distances from the pink and grey dots. If the authors want to use this theoretical model to describe their system, the predicted behavior should be in accordance with the experimental data in the whole range of the study and not only in the 5 μm plots.

- b) Regarding the influence of the results on the model used for the ballistic thermal conductivity, the authors have used Figure R3 to claim that their calculations are invariant respect to it. On the contrary, if one restricts on the temperature range where the authors are focusing their study, figure R3 shows exactly the contrary, that is, the slope in the ballistic conductivity is dependent on the model (see green and red lines in the attached figure). Additionally, the appearance of a maximum in the reduced thermal conductivity plot (Figure 4 of the main text) will depend on the position of the maximum on the ballistic thermal conductivity (red and green dots in figure R4).

- c) Comparing the plot of Figure 2 in the main article with the plots in Figure 5 of the Supplementary information it does not seem to be showing the emergence of any phenomena. I do not know the reason why the authors are not able to show it, but an observation of an increase of the thermal conductivity (even if it is small) should appear on that plot. This would be an evidence in favor of the authors claims.
- d) Regarding the theoretical framework used in the text to understand the hydrodynamic phenomena I still have some concerns. The authors still stick to the traditional idea that the abundance of normal collisions are necessary to have a hydrodynamic regime. The changes that they have included in the text respect to that is the inclusion of citations to the observation of second sound in kinetic materials but without connecting them with a hydrodynamic regime. If the claim of the necessity of normal collisions to obtain hydrodynamics is true, the model used to interpret the present results would be enough to show it. From the analysis of this report I have some doubts about this.

In my previous report I suggested to include alternative points of view that could lead to alternative descriptions of presented data. An alternative explanation of the results could be using a kinetic hydrodynamic model like the one used for silicon. Some of the authors of the paper are experts on this. The authors have their right of not to do it if they believe that these are irrelevant, but in that case they should give better scientific arguments in favor of their model.

Reviewer #2 (Remarks to the Author):

I cannot recommend the publication of this paper for a simple reason. The revised version still includes the following statement: "we demonstrate the phonon Poiseuille flow in a 5 μm -wide suspended graphite ribbon with purified ^{13}C isotope concentration, while not in the case with natural abundance. "

This binary statement about absence and presence of phonon Poiseuille flow in two different types of samples (natural vs. isotopically pure) is misleading. The experimental data is shown in Fig. 2. At 90K, the difference between the thermal conductivity of the types of samples is 20 percent. The data quality is such that error bars between the two sets of data overlap.

The new Fig. 5 tells the reader a very different story. According to it, the mean-free-path is ONE ORDER OF MAGNITUDE longer in the isotopically pure sample. However, what is presented in this figure is not data, but calculations, which are in flagrant contradiction with the raw data. Apparently, the contrast between the two has escaped the authors' attention and is not commented at all. If scattering by isotopes is so important, why should its experimental signature become as small as the experimental resolution?

Phonon hydrodynamics is attracting much attention. This paper makes a modest contribution to the field by demonstrating that isotopic purity enhances thermal conductivity of graphite ribbons by a modest value, comparable to what has been found in other solids like silicon or diamond. However, the present presentation of experimental facts is skewed in a way which makes it unsuitable for publication in a respectable journal.

Reviewer #3 (Remarks to the Author):

The authors have satisfactorily responded to my prior criticisms. I find the manuscript suitable for publication.

Reviewer #4 (Remarks to the Author):

All my concerns are addressed. I would recommend its publication in Nature Communications.

Reviewer #2 (Remarks to the Author):

I cannot recommend the publication of this paper for a simple reason. The revised version still includes the following statement: "we demonstrate the phonon Poiseuille flow in a 5 μm-wide suspended graphite ribbon with purified 13C isotope concentration, while not in the case with natural abundance. "

This binary statement about absence and presence of phonon Poiseuille flow in two different types of samples (natural vs. isotopically pure) is misleading. The experimental data is shown in Fig. 2. At 90K, the difference between the thermal conductivity of the types of samples is 20 percent. The data quality is such that error bars between the two sets of data overlap.

Response: We appreciate the time and effort of the reviewer in reading and evaluating our manuscript. The reviewer may have a misunderstanding about the main finding in the present work, as explained below:

- (1) Indeed, we obtained ~20% enhancement of thermal conductivity in purified graphite compared to the natural sample (more enhancement is expected when their widths are exactly the same in the absence of uncertainty from fabrication) as shown in Fig. 2. However, the thermal conductivity over the ballistic limit ($\kappa/G_{\text{ballistic}}$), as a criterion of phonon Poiseuille flow, shows qualitatively different trends of temperature dependence, as shown in Fig. 4. The increase of $\kappa/G_{\text{ballistic}}$ with increasing temperature in purified graphite ribbon definitely indicates the presence of phonon Poiseuille flow, in contrast to the continuous decrease of $\kappa/G_{\text{ballistic}}$ indicating its absence.
- (2) The error bars between the two data sets for natural and purified graphite ribbons only overlap at 90 K, whereas not more for others between 80-100 K.

The new Fig. 5 tells the reader a very different story. According to it, the mean-free-path is ONE ORDER OF MAGNITUDE longer in the isotopically pure sample. However, what is presented in this figure is not data, but calculations, which are in flagrant contradiction with the raw data. Apparently, the contrast between the two has escaped the authors' attention and is not commented at all. If scattering by isotopes is so important, why should its experimental signature become as small as the experimental resolution?

Response: Even though the mean-free-path of the isotope process is one order of magnitude longer in the purified sample compared to the natural one, the normal process is always dominant in both cases, as indicated by the sufficient short mean-free-path (or equivalently, large scattering rate) of the normal process in Fig. 5 of the main text. Hence, the theoretical calculation is not contradictive to our experimental data.

On the other hand, it also indicates that the occurrence of phonon Poiseuille flow is very sensitive to the resistive process. The normal process is around two and three orders of magnitude stronger than the resistive isotope scattering in natural and purified graphite ribbons, which results in the absence and presence of phonon Poiseuille flow, respectively, as consistent with the theoretical window condition ($l_N \ll W, l_R l_N \gg W^2$).

Phonon hydrodynamics is attracting much attention. This paper makes a modest contribution to the field by demonstrating that isotopic purity enhances thermal conductivity of graphite ribbons by a modest value, comparable to what has been found in other solids like silicon or diamond. However, the present presentation of experimental facts is skewed in a way which makes it unsuitable for publication in a respectable journal.

Response: Besides the quantitative enhancement of the thermal conductivity in graphite by isotopic purification, the importance of our contribution is to demonstrate *a qualitatively different* temperature scaling of $\kappa/G_{\text{ballistic}}$ in graphite ribbons with different widths and isotope concentrations. Our work theoretically reveals the criterion for evidencing the phonon Poiseuille flow in graphite and further confirms and observes this phenomenon from an experimental point of view.

As a result, the current manuscript delivers an important message to the physics, carbon and heat transfer community that the experimental observation of phonon Poiseuille flow is much more challenging than this community realizes. Both the isotopic enrichment and proper structure design are indispensable for the observation of phonon Poiseuille flow, which is crucial in future attempts to make phonon hydrodynamics practical towards room temperatures.

Reviewers' comments:

Reviewer #1 (Remarks to the Author):

The authors have not answered to any of the questions of my previous review. There I pointed to numerous questions regarding the statistical analysis of the data and the theoretical interpretation of the results.

I want to recognise the quality of the work done. I am aware about the extreme difficulty of the measurements that are here presenting. I also want to point the importance of the study that here is done. Despite of this, I would like the authors to notice that the doubts presented by all the referees in the submission are going in the same direction. They don't see a clear experimental and theoretical evidence in favor of the conclusions.

The question about the presence or not of Poiseuille flow and second sound has adquired in the last years a lot of new arguments respect to the simple normal phonon-phonon scattering arguments of the 60s. There are now some works pointing towards the problem that the behaviour of the phonons near the zone center can be problematic. There are also works showing hydrodynamic behavior in other materials like black phosphorous, germanium or silicon.

All these works can give a new insight to the results that the authors are presenting, but they have decided not to use it in their study. I firmly believe that an analysis including some of the new arguments that the topic has today will clearly give a better quality to the work.

The discussion in this paper is still based in the 60's theory. From my point of view, the theoretical framework presented in the article is partial and not connected with the state of the art. Despite the authors are of the respectable opinion that all these recent publications are wrong, a paper with the quality to be published in Nature Communications should include a proper discussion on it.

For all these reasons I believe that the paper does not deserve publication in Nature Communications.

Reviewer #2 (Remarks to the Author):

The authors have not altered their conclusion. The abstract declares:

"we demonstrate the phonon Poiseuille flow in a 5 μm -wide suspended graphite ribbon with purified ^{13}C isotope concentration, while not in the case with natural abundance."

What is the basis of this statement?

Fig. 2 of this paper shows the RAW data. The difference between the data for the two samples is barely larger than the experimental margin of error.

Fig. 4 shows the COOKED data. the purified sample shows a local peak, while the natural sample does not. But to this reviewer, both sets of data contain shoulders, admittedly more visible in the purified sample. But where are the peaks of the theoretical solid lines? One needs a fair dose of wishful thinking to invoke a QUALITATIVE difference.

I remain unconvinced of the solidity of the conclusion.

Reviewer #3 (Remarks to the Author):

I find that the authors' revisions have addressed my initial concerns and have improved the manuscript. The addition of data and analysis for Si specimens (demonstrating the absence of Poiseuille flow as anticipated given weaker normal scattering) have strengthened the authors' conclusions regarding Poiseuille flow in graphite. Combined with the introduction of what appears to be a more apt criterion for assessing Poiseuille flow of phonons in anisotropic compounds, I find the manuscript suitable for publication.

Reviewer #4 (Remarks to the Author):

My comments are addressed, and I do not have further comments

Response Letter for Observation of phonon Poiseuille flow in isotopically-purified graphite ribbons

Xin Huang^{1,†}, Yangyu Guo^{1,†}, Yunhui Wu¹, Satoru Masubuchi¹, Kenji Watanabe², Takashi Taniguchi^{1,3}, Zhongwei Zhang¹, Sebastian Volz^{1,4}, Tomoki Machida¹, and Masahiro Nomura^{1,5,*}

¹Institute of Industrial Science, The University of Tokyo, Tokyo 153-8505, Japan

²Research Center for Functional Materials, National Institute for Materials Science, Tsukuba 305-0044, Japan

³International Center for Materials Nanoarchitectonics, National Institute for Materials Science, Tsukuba 305-0044, Japan

⁴LIMMS, CNRS-IIS IRL 2820, The University of Tokyo, Tokyo 153-8505, Japan

⁵Research Center for Advanced Science and Technology, The University of Tokyo, Tokyo 153-0041, Japan

*corresponding author: Masahiro Nomura (nomura@iis.u-tokyo.ac.jp)

†these authors contributed equally to this work

Reviewer #1 (Remarks to the Author):

The authors have not answered to any of the questions of my previous review. There I pointed to numerous questions regarding the statistical analysis of the data and the theoretical interpretation of the results.

Response: First of all, please accept our sincerest apologies that we could not response to the reviewer's comments without much improved results at that moment due to the deadline. We appreciate your careful and considered valuable remark. We have taken your points seriously, and **conducted additional experiments and corresponding theoretical calculations** in this half year. Also we added relevant discussions about the modern viewpoint of the hydrodynamic regime. A detailed point-by-point response to reviewer's comments in both 2nd- and 3rd-round reviews is provided as follows.

Response to 2nd-round review:

I have included a pdf report as it is mostly based on the analysis of the figures in the article

Response: We appreciate the reviewer for his/her time in critical reading of our work and the pertinent comments and suggestions for us. We have carefully conducted new experiments with better results, and improved our manuscript correspondingly in accordance with your advice, as explained below:

The authors have not answered the questions and doubts regarding my previous report.

a) Regarding the questions about the statistical significance of the increasing in thermal conductivity respect the ballistic conductivity, the authors have limited to include a zoomed plot with the data from 30 to 50 K.

1. When I pointed to the **statistical** significance of the maximum, I was not indicating that I didn't see an increasing trend between that temperature ranges (that was obvious). My concern was about that there is not a clear work to show that this trend was not because of experimental error. I explain this point in order to clarify it.

Looking at the image on the left, the justification in favour of a Poiseuille flow is based in the fact that in region 1 there is a maximum in the normalized thermal conductivity. In order for this to be **statistically acceptable**, the authors should include some argumentation about the minimum at region 2 and a new local maximum in region 3. From my point of view, these are showing the large experimental error of the measure. Without the proper description of these fluctuations, the argument that in region 1 we are observing the emergence of a new phenomenon is statistically loose.

Response: Due to the extremely strict window condition, the phonon Poiseuille flow might be destructed by unknown defects or contaminations during sample fabrication process which introduces additional resistive scattering. The sample conditions vary among different graphite flakes, and should be carefully controlled in the actual fabrications. In the new experiment, we fabricated 30 μm -long, 85 nm-thick isotopically purified graphite ribbons with the width of 1.3, 3.3 and 5.5 μm . We did our best to keep the quality of the samples as high as possible, and obtained more statistically acceptable experimental data.

As seen in Fig. 2R1, in the purified graphite ribbon of 5.5 μm width, we observed the increase of normalized thermal conductivity ($\kappa/G_{\text{ballistic}}$) from ~ 40 K to ~ 90 K. It indicates definite evidence of phonon Poiseuille flow even considering the statistical uncertainties.

Fig. 2R1. Thermal conductivity over ballistic thermal conductance ($\kappa/G_{\text{ballistic}}$) as a function of temperature. The dark (light) blue dots represent the experimental data of isotopically-purified (natural) graphite ribbons with a designed width of 5.5 μm .

I am aware that the authors are in the limit of resolution of a very challenging measure. In that case, they have an alternative to solve this problem. They can use a theoretical model where the predictions demonstrate the appearance of this emergent phenomena in the same place where the data seems to be indicating it. In that case, the presented model (based on the BTE) does not seem to be indicating this. In the following plot, the model (solid lines) is predicting an increase in the reduced thermal conductivity in both regions (1 and 2).

The authors suggestion that the increase of thermal conductivity in region 2 is a clear evidence of Poiseuille should be accompanied with a model where in region 1 is not predicting this increase.

Response: As mentioned by the reviewer, the observation of phonon Poiseuille flow is indeed a challenging task, especially, in the graphite ribbon with an intermediate width, i.e., $\sim 3 \mu\text{m}$ wide ribbon, where the hydrodynamic effect is relatively weak and easily to be destroyed by any unknown defects or contaminations.

However, in the new experimental results, we observed clear evidence of phonon Poiseuille flow in the purified graphite ribbon of $3.3 \mu\text{m}$ width, i.e., an increase of $\kappa/G_{\text{ballistic}}$ from $\sim 50 \text{ K}$ to $\sim 80 \text{ K}$, as seen in Fig. 2R2a. In the $5.5 \mu\text{m}$ -wide ribbon, stronger phonon Poiseuille flow was observed, as shown by the more prominent increase of $\kappa/G_{\text{ballistic}}$ with temperature, as illustrated in Fig. 2R2b.

Fig. 2R2. Thermal conductivity over ballistic thermal conductance ($\kappa/G_{\text{ballistic}}$) as a function of temperature. The red (pink) and dark (light) blue dots represent the experimental data of isotopically-purified (natural) graphite ribbons with the designed widths of (a, c) $3.3 \mu\text{m}$ and (b, d) $5.5 \mu\text{m}$, respectively. The empty dots with the corresponding colors denote the modelling results by BTE with first-principles inputs.

The experimental tendencies are generally reproduced by our modeling results based on a direct solution of phonon Boltzmann transport equation (BTE) with fully first-principles inputs, as shown in Fig. 2R2c, d. There is some difference in the absolute values of $\kappa/G_{\text{ballistic}}$, since an infinite thickness (equivalent to ideally smooth surfaces in c -axis) is considered in our modeling. A direct solution of phonon BTE for hydrodynamic heat transport in graphite ribbon with finite length, width and thickness remains very challenging, as it requires a numerical solution in both 3D reciprocal space and 3D real space. However, a preferable consistence is found in terms of: 1) the qualitative trends of isotopically-purified and natural abundance graphite ribbons and 2) the temperatures where the minimum and maximum of $\kappa/G_{\text{ballistic}}$ emerge.

2. In point 2, the authors use again the zoom to evidence the change in the trend of their data in the 30-50 K range. This does not respond my concern about the lack of predictability of the model they are using.

As it can be seen in the figure on the right, the black arrows and red arrows show a clear tendency in the model where the pure and natural abundance samples split when increasing its size. This is clearly not observed in the experimental data. In

my previous report I combined the plots to show this. In this figure it can be observed that the red and black dots (data in the ellipse) are at similar distances from the pink and grey dots. If the authors want to use this theoretical model to describe their system, the predicted behavior should be in accordance with the experimental data in the whole range of the study and not only in the 5 μm plots.

Response: We appreciate this crucial point from the reviewer. In our new samples, the splitting of $\kappa/G_{\text{ballistic}}$ of purified and natural graphite ribbons is clearly observed in experiments when increasing the sample width. As shown in Fig. 2R3, the more pronounced difference of $\kappa/G_{\text{ballistic}}$ between purified and natural graphite samples is clearly seen with the enlargement of ribbon width from 1.3 μm to 3.3 μm , attributed to the larger space for the normal scattering to occur in wider ribbons. This trend is indeed consistent with that predicted by our theoretical modelling.

Fig. 2R3. Thermal conductivity over ballistic thermal conductance ($\kappa/G_{\text{ballistic}}$) as a function of temperature. The dark (light) green and red (pink) dots represent the experimental data of isotopically-purified (natural) graphite ribbons with the designed widths of (a, c) 1.3 μm and (b, d) 3.3 μm , respectively. The empty dots with the corresponding colors denote the modelling results by BTE with first-principles inputs.

To answer the reviewer's concerns in question a), we added the following discussions in the main text (page 7, lines 176-190) and Methods (page 10, lines 314-323) of the revised manuscript:

“The aforementioned tendencies of experimental data are generally consistent with our theoretical modelling results in Fig. 4g-i based on a direct solution of phonon Boltzmann transport equation (BTE) with full first-principles inputs (see details in Methods). There is some difference between the absolute values of $\kappa/G_{\text{ballistic}}$ in experimental and theoretical results, as we consider infinite thickness in the BTE modeling. However, a good agreement is found in terms of the relative trends of isotopically-purified and natural abundance graphite ribbons and the temperatures where the minimum and maximum emerge. A direct solution of phonon BTE for hydrodynamic heat transport in graphite ribbon with finite length, width and thickness is a challenging task, as it requires a numerical solution in both 3D reciprocal space and 3D real space. To the authors' best knowledge, it is only reported that the Monte Carlo solution of phonon BTE with ab initio full scattering term for such situation from one group in very recent studies^{15,55}. However, due to huge computational cost, relatively coarse grids in both reciprocal and real spaces have been adopted. Apparently, there is still some space to further improve the accuracy of the numerical solution and its agreement with experimental result¹⁵. On the other hand, as the thickness effect on basal-plane heat transport in graphite remains an open question^{35,55}, the present modeling and experimental study are mainly focused on the effects of finite length and width. Our semi-quantitative theoretical modeling generally provides a good guide for the observation of phonon Poiseuille flow in finite-sized isotopically purified graphite ribbons.”

“Along the cross-plane direction of graphite ribbon, we assume infinite thickness and solve the phonon BTE in 2D real space and 3D reciprocal space³³. Such treatment is strictly valid when the surfaces in the thickness direction are fully specular, i.e., ideally smooth, which is an acceptable case when the sample is obtained via perfect exfoliation process as in multilayer graphene ribbon^{74,75}. Since the absolute thermal conductivity by the present BTE modeling is larger than the experimental one, we infer that there should be unknown surface imperfections in the thickness direction. Nevertheless, the present BTE modeling captures the dominant effects from finite length and width, considering that the numerical solution of BTE in both 3D real and reciprocal spaces is computationally too expensive for hydrodynamic heat transport. Also, our recent numerical methodology considering only finite length and width³³ remains to be further developed, which requires appreciable amount of future work and effort.”

Furthermore, to provide a clearer demonstration of the width- and isotope-dependence of the phonon Poiseuille flow, we replaced Fig. 4 in the manuscript by Fig. 2R4 below.

Fig. 2R4. (a-c) SEM images of suspended isotopically-purified graphite ribbons with the widths of 1.3 μm , 3.3 μm and 5.5 μm , respectively. (d-f) Experimentally measured and (g-i) calculated thermal conductivity over ballistic thermal conductance ($\kappa/G_{\text{ballistic}}$) as a function of temperature corresponding to the three ribbons in (a-c). The dark (light) green, red (pink), and dark (light) blue dots represent the experimental data of isotopically-purified (natural) graphite ribbons. The empty dots with the corresponding colors denote the modelling results by BTE with first-principles inputs. Note that the actual widths of the natural graphite ribbons are 1.6 μm , 3.7 μm and 6.3 μm , respectively, due to the deviation in fabrication, resulting in the minor flip of thermal conductivities at very low temperatures.

b) Regarding the influence of the results on the model used for the ballistic thermal conductivity, the authors have used Figure R3 to claim that their calculations are invariant respect to it. On the contrary, if one restricts on the temperature range where the authors are focusing their study, figure R3 shows exactly the contrary, that is, the slope in the ballistic conductivity is dependent on the model (see green and red lines in the attached figure). Additionally, the appearance of a maximum in the reduced thermal conductivity plot (Figure 4 of the main text) will depend on the position of the maximum on the ballistic thermal conductivity (red and green dots in figure R4).

Response: The reviewer is correct. Indeed the quantitative value of ballistic thermal conductance ($G_{\text{ballistic}}$) and the peak position of $G_{\text{ballistic}}/T^{2.5}$ will be dependent on the atomic interaction potential. If we compare the slopes of $G_{\text{ballistic}}/T^{2.5}$ by the empirical potential and by DFT at the same temperature, it will be not exactly the same, as shown by the reviewer. However, the temperature-scaling of $G_{\text{ballistic}}/T^{2.5}$ below the peak, i.e., the ballistic limit, is almost the same. Indeed, even if we adopt the $G_{\text{ballistic}}$ calculated by the empirical potential, our concluding remarks about the evidence of phonon Poiseuille flow will not be changed, as shown in Fig. 2R5 below for the 5.5 μm -wide isotopically purified graphite ribbon. Since the ballistic thermal conductance of graphite does not have a trivial temperature dependence as in 3D isotropic crystals, an accurate description of it will be crucial for evidencing the occurrence of phonon Poiseuille flow. Therefore, we still recommend to use the DFT calculation to obtain the ballistic thermal conductance [as in a recent study J. Appl. Phys. **131**, 075104

(2022)], or empirical potential which well captures the dispersion of low-lying phonon branches that dominate the contribution at low temperatures [as in a previous study Nat. Commun. 4, 1734 (2013)].

Fig. 2R5. Thermal conductivity (κ) over ballistic thermal conductance ($G_{\text{ballistic}}$) of isotopically purified graphite ribbon with 5.5 μm width. The ballistic thermal conductance is obtained based on DFT calculation and empirical interatomic potential respectively.

c) Comparing the plot of Figure 2 in the main article with the plots in Figure 5 of the Supplementary information it does not seem to be showing the emergence of any phenomena. I do not know the reason why the authors are not able to show it, but an observation of an increase of the thermal conductivity (even if it is small) should appear on that plot. This would be an evidence in favor of the authors' claims.

Response: In the new samples, we observed a clear increase of the thermal conductivity from the natural to isotopically purified graphite, especially in the intermediate temperature range (i.e., 50-100 K), as shown in Fig. 2 for the 5.5 μm width in the main text (Fig. 2R6 below) and in Supplementary Fig. 3 for the 1.3 μm and 3.3 μm width (Fig. 2R7 below). Also note that the plot of thermal conductivity is in log-scale, which visually reduces the difference. For instance, the thermal conductivity of isotopically purified graphite ribbon with 5.5 μm width is enhanced by 105% compared to that of the natural one at 90 K.

Fig. 2R6. Temperature-dependent in-plane thermal conductivity of isotopically-purified and natural graphite ribbons with a designed width of 5.5 μm . The dark and light blue dots represent the results of isotopically-purified (0.02% ^{13}C) and natural (1.1% ^{13}C) graphite ribbons, respectively. Note that the actual width of the natural graphite ribbon is 0.8 μm wider than that of the isotopically-purified one due to the deviation in fabrication, resulting in the minor flip of thermal conductivities at very low temperatures. Inset: thermal conductivity data from 50 to 100 K.

Fig. 2R7. Thermal conductivity as a function of temperature for the isotopically-purified graphite ribbons with the designed widths of (a) 1.3 μm and (b) 3.3 μm . The dark (light) green and red (pink) dots represent the results of isotopically-purified (natural) graphite ribbons. Note that the actual widths of these two natural graphite ribbons are 0.3 μm and 0.4 μm wider than that of the isotopically-purified ones due to the deviation in fabrication, resulting in the minor flip of thermal conductivities at very low temperatures. Inset: thermal conductivity data in normal scale from 50 to 100 K.

d) Regarding the theoretical framework used in the text to understand the hydrodynamic phenomena I still have some concerns. The authors still stick to the traditional idea that the abundance of normal collisions are necessary to have a hydrodynamic regime. The changes that they have included in the text respect to that is the inclusion of citations to the observation of second sound in kinetic materials but without connecting them with a hydrodynamic regime. If the claim of the necessity of normal collisions to obtain hydrodynamics is true, the model used to interpret the present results would be enough to show it. From the analysis of this report I have some doubts about this.

In my previous report I suggested to include alternative points of view that could lead to alternative descriptions of presented data. An alternative explanation of the results could be using a kinetic hydrodynamic model like the one used for silicon. Some of the authors of the paper are experts on this. The authors have their right of not to do it if they believe that these are irrelevant, but in that case they should give better scientific arguments in favor of their model.

Response: We appreciate the reviewer for this excellent proposal. Indeed macroscopic phonon hydrodynamic models [for instance, the kinetic-collective model (KCM) in Phys. Rev. Mater. **2**, 076001 (2018)] will be able to describe the Poiseuille flow of phonons discussed in this work. The KCM covers both the nanoscale heat transport in kinetic materials like silicon, and the collective heat transport in graphitic materials where normal scattering dominates. In this work, we are focused on the latter, i.e., the collective limit. However, we are not denying the possibility of hydrodynamic regime in kinetic materials. Due to the nonlinear phonon properties and the complicated 3D geometries of the anisotropic graphite ribbons, it remains a non-trivial task to employ a hydrodynamic model [for instance, the most recent one in Phys. Rev. B **103**, L140301 (2021)] for the theoretical description in this manuscript. This is the reason why we only use a direct solution of phonon BTE, which represents a common theoretical ground although with higher computational cost. Definitely, a hydrodynamic description of the phonon Poiseuille flow is an efficient way with also a clear physical picture, which is an on-going effort and future target of the authors. For that, we add a paragraph to discuss the modern viewpoint about the phonon hydrodynamics at the end of the Discussion part (page 8-9, lines 232-246) in the revised version as follows:

“Finally, we would like to note that the phonon Poiseuille flow could be modeled by macroscopic phonon hydrodynamic equations^{2,56} in a much more efficient way. It resembles the description of Poiseuille flow of classical fluids by the Navier-Stokes equation. The classical Guyer-Krumhansl (G-K) phonon hydrodynamic equation⁵⁶ assumes gray phonon properties and usually works well for traditional 3D isotropic crystals at very low temperatures. However, the complex nonlinear frequency-dependent phonon properties are important for phonon hydrodynamics in graphitic materials where the original G-K equation might be not able to be directly applied. Recently, a generalized G-K heat transport equation (so-called kinetic-collective model, KCM⁶²) has been derived from phonon BTE taking into account the arbitrary scattering term and the nonlinear phonon properties⁶³. In principle, the KCM, together with appropriate macroscopic boundary conditions, could be an alternative for modeling the phonon Poiseuille flow in finite-sized graphite ribbon, which is however a nontrivial task and beyond the scope of the present study. On the other hand, the hydrodynamic model (or KCM) covers the heat transport in both kinetic^{62,64} and collective⁶⁵ limits. Apparently the debate about the definition of the hydrodynamic regime^{14,19} when heat transport could be described by the KCM is still open. Nevertheless, the present work is mainly focused on the collective limit, i.e. when normal scattering dominates. Thus we could not provide definite remarks about this debate in the current stage, and leave it for a future exploration.”

Response to 3rd-round review:

I want to recognise the quality of the work done. I am aware about the extreme difficulty of the measurements that are here presenting. I also want to point the importance of the study that here is done. Despite of this, I would like the authors to notice that the doubts presented by all the referees in the submission are going in the same direction. They don't see a clear experimental and theoretical evidence in favor of the conclusions.

The question about the presence or not of Poiseuille flow and second sound has acquired in the last years a lot of new arguments respect to the simple normal phonon-phonon scattering arguments of the 60s. There are now some works pointing towards the problem that the behaviour of the phonons near the zone center can be problematic. There are also works showing hydrodynamic behavior in other materials like black phosphorous, germanium or silicon.

All these works can give a new insight to the results that the authors are presenting, but they have decided not to use it in their study. I firmly believe that an analysis including some of the new arguments that the topic has today will clearly give a better quality to the work.

The discussion in this paper is still based in the 60's theory. From my point of view, the theoretical framework presented in the article is partial and not connected with the state of the art. Despite the authors are of the respectable opinion that all these recent publications are wrong, a paper with the quality to be published in Nature Communications should include a proper discussion on it.

For all these reasons I believe that the paper does not deserve publication in Nature Communications.

Response: We appreciate the reviewer's time in evaluating our work, and very pertinent comments for us to improve our manuscript. To demonstrate a clearer evidence of phonon Poiseuille flow, we have

carefully conducted additional experiments with better results and corresponding theoretical calculations. In addition, we have added discussions about the theoretical description of the phonon Poiseuille flow by kinetic-collective hydrodynamic models (KCM). As we explained, the present work is focused on the heat transport in the collective limit, and aims to demonstrate the first unambiguous evidence of phonon Poiseuille flow in graphitic materials. We are not denying the recent opinion of hydrodynamic regime in kinetic materials like Si and Ge, since the KCM covers both the kinetic limit and the collective limit studied here. Actually, the definition of hydrodynamic regime remains still an open debate in the community, as known by the reviewer. Due to the difficulty in the implementation of the KCM in 3D anisotropic graphite ribbons, we only use the phonon BTE which could in principle capture heat transport in both limits in the sacrifice of some efficiency. We hope that our response answers the reviewer's concerns, and the reviewer will find that the revised manuscript deserves publication in Nature Communications.

Reviewer #2 (Remarks to the Author):

The authors have not altered their conclusion. The abstract declares:

"we demonstrate the phonon Poiseuille flow in a 5 μm -wide suspended graphite ribbon with purified ^{13}C isotope concentration, while not in the case with natural abundance."

What is the basis of this statement?

Response: We appreciate the reviewer's comments for us to improve our manuscript. To answer the reviewer's concerns on this work, we conducted additional experiments using carefully prepared graphite samples with high quality and strengthened our conclusions with much improved results, as explained below. In addition, we changed our declaration in the abstract of the revised manuscript in accordance with your suggestion. We removed the statement of "*while not in the case with natural abundance.*" and altered our conclusion as follow:

"we demonstrate the existence of the phonon Poiseuille flow in a 5.5 μm -wide, suspended and isotopically-purified graphite ribbon up to a temperature of 90 K."

Fig. 2 of this paper shows the RAW data. The difference between the data for the two samples is barely larger than the experimental margin of error.

Response: Due to the extremely strict window condition, the phonon Poiseuille flow might be destructed by unknown defects or contaminations during sample fabrication process which introduces additional resistive scattering. The sample conditions vary among different graphite flakes, and should be carefully controlled in the actual fabrications. In the new experiment, we fabricated 30 μm -long and 85 nm-thick purified graphite ribbons with various widths. We did our best to keep the quality of the samples as high as possible, and obtained better experimental data. In Fig. 3R1, we compared the thermal conductivity of isotopically-purified (0.02% ^{13}C) and natural (1.1% ^{13}C) graphite ribbons with a designed width of 5.5 μm from 10 K to room temperature. A clear isotope effect on the thermal conductivity is demonstrated in the intermediate temperature range (as further enlarged in the inset). The thermal conductivity of the purified sample (1235 W/mK) is more than twice that of the natural one (633 W/mK) at 90 K. The difference between the data for the two samples is smaller than the experimental margin of error.

Fig. 3R1. Temperature-dependent in-plane thermal conductivity of isotopically-purified and natural graphite ribbons with a designed width of 5.5 μm . The dark and light blue dots represent the results of isotopically-purified (0.02% ^{13}C) and natural (1.1% ^{13}C) graphite ribbons, respectively. Note that the actual width of the natural graphite ribbon is 0.8 μm wider than that of the isotopically-purified one due to the deviation in fabrication, resulting in the minor flip of thermal conductivities at very low temperatures. Inset: thermal conductivity data from 50 to 100 K.

To answer the reviewer's concerns, we replaced Fig. 2 in the manuscript by Fig. 3R1, and updated the relevant discussions in the main text (page 3, lines 84-92).

Fig. 4 shows the COOKED data. the purified sample shows a local peak, while the natural sample does not. But to this reviewer, both sets of data contain shoulders, admittedly more visible in the purified sample. But where are the peaks of the theoretical solid lines? One needs a fair dose of wishful thinking to invoke a QUALITATIVE difference.

I remain unconvinced of the solidity of the conclusion.

Response: We thank the reviewer for the comments for us to improve this crucial point. With our improved experimental results, we fairly studied the temperature dependence of absolute values of $\kappa/G_{\text{ballistic}}$ for graphite ribbons with varying widths and isotope contents, as shown in Fig. 3R2. We observed a clear local peak of $\kappa/G_{\text{ballistic}}$ in the purified graphite ribbons of 3.3 μm and 5.5 μm widths in Fig. 3R2e and Fig. 3R2f, respectively. In contrast, as the temperature increases, $\kappa/G_{\text{ballistic}}$ in the natural abundance graphite ribbons of similar widths generally decreases monotonously within experimental uncertainties. Such difference of the trend indicates the superballistic heat transport in the isotopically purified graphite, i.e., the evidence of phonon Poiseuille flow, as the crucial contribution of the present work.

Fig. 3R2. (a-c) SEM images of suspended isotopically-purified graphite ribbons with the widths of 1.3 μm , 3.3 μm and 5.5 μm , respectively. (d-f) Experimentally measured and (g-i) calculated thermal conductivity over ballistic thermal conductance ($\kappa/G_{\text{ballistic}}$) as a function of temperature corresponding to the three ribbons in (a-c). The dark (light) green, red (pink), and dark (light) blue dots represent the experimental data of isotopically-purified (natural) graphite ribbons. The empty dots with the corresponding colors denote the modelling results by BTE with first-principles inputs. Note that the actual widths of the natural graphite ribbons are 1.6 μm , 3.7 μm and 6.3 μm , respectively, due to the deviation in fabrication, resulting in the minor flip of thermal conductivities at very low temperatures.

The observed trend is generally well captured by our BTE modeling of graphite ribbons with the same length, width and isotope content as those in experiment. An accurate solution of phonon BTE for hydrodynamic heat transport in graphite ribbon with finite length, width and thickness remains a challenging task, as it requires a numerical solution in both 3D reciprocal space and 3D real space. In addition, the thickness effect on the basal-plane heat transport along graphite is still an open question in the community. Thus an infinite thickness is considered in our theoretical modeling, which also refers to perfectly smooth surfaces in the c -axis. In actual experiments, there should exist unknown imperfections (roughness or contaminations) on the sample surfaces, which explains the difference of the absolute values of $\kappa/G_{\text{ballistic}}$ between the modeling and observed results. However, a preferable agreement is still found between the experiment and modeling in terms of: 1) the qualitative trends of isotopically-purified and natural graphite ribbons and 2) the temperatures where the minimum and maximum of $\kappa/G_{\text{ballistic}}$ emerge, as shown in Fig. 3R2(g-i).

To clarify the evidence for the occurrence of phonon Poiseuille flow and provide clearer width- and isotope-dependence, we replaced Fig. 4 in the manuscript by Fig. 3R2.

We thank the reviewer again for all the valuable comments and suggestions. Hope that our point-by-point response may answer the reviewer's concerns and the improved manuscript is suitable for publication.

Reviewer #3 (Remarks to the Author):

I find that the authors' revisions have addressed my initial concerns and have improved the manuscript. The addition of data and analysis for Si specimens (demonstrating the absence of Poiseuille flow as anticipated given weaker normal scattering) have strengthened the authors' conclusions regarding Poiseuille flow in graphite. Combined with the introduction of what appears to be a more apt criterion for assessing Poiseuille flow of phonons in anisotropic compounds, I find the manuscript suitable for publication.

Response: We appreciate the reviewer for the recognition of the importance of our work and the agreement of publication in Nature Communications. We thank the reviewer again for all the comments and suggestions to enrich the contents of our manuscript.

Reviewer #4 (Remarks to the Author):

My comments are addressed, and I do not have further comments

Response: We appreciate the reviewer for the recognition of the importance of our work and the agreement of publication in Nature Communications. We thank the reviewer again for all the comments and suggestions to improve our manuscript.

REVIEWERS' COMMENTS

Reviewer #1 (Remarks to the Author):

The authors have significantly improved their work with the new data they have included. With the new data, the emergence of a new trend in is now observed beyond the statistical fluctuations.

Due to this effort, my concerns about the conclusions have been resolved and my opinion is that the work now has the quality standards to be published in this journal.

I want to recognize the efforts done by the authors in order to response to my many questions and also the extraordinary work that they have done in this review process.

Reviewer #2 (Remarks to the Author):

What I found contestable in the previous version of the paper was the claim that there is a qualitative difference between ordinary and isotopically pure graphite. Such a qualitative difference was undetectable in the raw data, making such a claim misleading.

The authors have amended their paper. Their raw data (Fig. 2) is more convincing for claiming that here isotopically purity matters more than in the case of, say silicon. Their formulation of the main finding is also more subtle and less contestable.

Looking at their new Fig. 3 and Fig. 4, I think that their claim is reasonable: In their isotopically pure samples in contrast to their natural samples, the presence of Poiseuille flow appears to be beyond reasonable doubt. I recommend the publication of this version.

Reviewer #3 (Remarks to the Author):

Though I previously indicated support for publication in a prior round of review, I have read through the author responses to the criticism of other reviewers -- my opinion remains unchanged.

I would comment that I am a little surprised that the new data (e.g. presented in Fig.'s 2R4, 2R7, 3R1) is fairly coarsely spaced at $T > 100$ K where some differences in thermal conductivity between specimens differing in isotopic purity are being compared with the intent to better address referee concerns about precisely such differences.

Reviewer #4 (Remarks to the Author):

My comments are addressed, and I do not have further comments

Response Letter for Observation of phonon Poiseuille flow in isotopically purified graphite ribbons

Xin Huang^{1,†}, Yangyu Guo^{1,†}, Yunhui Wu¹, Satoru Masubuchi¹, Kenji Watanabe², Takashi Taniguchi^{1,3}, Zhongwei Zhang¹, Sebastian Volz^{1,4}, Tomoki Machida¹, and Masahiro Nomura^{1,5,*}

¹Institute of Industrial Science, The University of Tokyo, Tokyo 153-8505, Japan

²Research Center for Functional Materials, National Institute for Materials Science, Tsukuba 305-0044, Japan

³International Center for Materials Nanoarchitectonics, National Institute for Materials Science, Tsukuba 305-0044, Japan

⁴LIMMS, CNRS-IIS IRL 2820, The University of Tokyo, Tokyo 153-8505, Japan

⁵Research Center for Advanced Science and Technology, The University of Tokyo, Tokyo 153-0041, Japan

*corresponding author: Masahiro Nomura (nomura@iis.u-tokyo.ac.jp)

†these authors contributed equally to this work

Reviewer #1 (Remarks to the Author):

The authors have significantly improved their work with the new data they have included. With the new data, the emergence of a new trend in is now observed beyond the statistical fluctuations.

Due to this effort, my concerns about the conclusions have been resolved and my opinion is that the work now has the quality standards to be published in this journal.

I want to recognize the efforts done by the authors in order to response to my many questions and also the extraordinary work that they have done in this review process.

Response: We appreciate the reviewer for the recognition of the importance of our work and the agreement to publish in Nature Communications. We thank the reviewer again for all the comments and suggestions to improve the quality of our manuscript.

Reviewer #2 (Remarks to the Author):

What I found contestable in the previous version of the paper was the claim that there is a qualitative difference between ordinary and isotopically pure graphite. Such a qualitative difference was undetectable in the raw data, making such a claim misleading.

The authors have amended their paper. Their raw data (Fig. 2) is more convincing for claiming that here isotopically purity matters more than in the case of, say silicon. Their formulation of the main finding is also more subtle and less contestable.

Looking at their new Fig. 3 and Fig. 4, I think that their claim is reasonable: In their isotopically pure samples in contrast to their natural samples, the presence of Poiseuille flow appears to be beyond reasonable doubt. I recommend the publication of this version.

Response: We appreciate the reviewer for the agreement to publish in Nature Communications. We thank the reviewer again for all the comments and suggestions to solidify our conclusions and the recognition of the amended version of our manuscript.

Reviewer #3 (Remarks to the Author):

Though I previously indicated support for publication in a prior round of review, I have read through the author responses to the criticism of other reviewers -- my opinion remains unchanged.

I would comment that I am a little surprised that the new data (e.g. presented in Fig.'s 2R4, 2R7, 3R1) is fairly coarsely spaced at $T > 100$ K where some differences in thermal conductivity between specimens differing in isotopic purity are being compared with the intent to better address referee concerns about precisely such differences.

Response: We appreciate the reviewer's time in reading our responses to the other reviewers and the agreement to publish in Nature Communications. We thank the reviewer for the recognition of our new results and all the comments and suggestions to enrich the contents of our manuscript.

Reviewer #4 (Remarks to the Author):

My comments are addressed, and I do not have further comments

Response: We appreciate the reviewer for evaluating our work and the agreement to publish in Nature Communications. We thank the reviewer again for all the comments and suggestions to improve our manuscript.